



# $\delta^{11}$B as monitor of calcification site pH in divergent marine calcifying organisms

**Jill N. Sutton**[1], **Yi-Wei Liu**[1,2], **Justin B. Ries**[3], **Maxence Guillermic**[2], **Emmanuel Ponzevera**[4], **and Robert A. Eagle**[1,5,6]

[1]Université de Brest, UMR 6539 CNRS/UBO/IRD/Ifremer, LEMAR, IUEM, 29280, Plouzané, France
[2]Université de Brest, UMR 6539 CNRS/UBO/IRD/Ifremer, LGO, IUEM, 29280, Plouzané, France
[3]Department of Marine and Environmental Sciences, Marine Science Center, Northeastern University,
430 Nahant Rd, Nahant, MA 01908, USA
[4]Unité de Recherche Géosciences Marines, Ifremer, 29280, Plouzané, France
[5]Institute of the Environment and Sustainability, University of California, Los Angeles, LaKretz Hall,
619 Charles E Young Dr E no. 300, Los Angeles, CA 90024, USA
[6]Atmospheric and Oceanic Sciences Department, University of California, Los Angeles, Maths Science CE1 Building,
520 Portola Plaza, Los Angeles, CA 90095, USA

**Correspondence:** Jill N. Sutton (jill.sutton@univ-brest.fr) TS1 and Robert A. Eagle (robeagle@g.ucla.edu)

**Abstract.** TS2 TS3 The boron isotope composition ($\delta^{11}$B) of marine biogenic carbonates has been predominantly studied as a proxy for monitoring past changes in seawater pH and carbonate chemistry. However, a number of assumptions regarding chemical kinetics and thermodynamic isotope exchange reactions are required to derive seawater pH from $\delta^{11}$B from biogenic carbonates. It is also probable that $\delta^{11}$B of biogenic carbonate reflects seawater pH at the organism's site of calcification, which may or may not reflect seawater pH. Here, we report the development of methodology for measuring the $\delta^{11}$B of biogenic carbonate samples at the multi-collector inductively coupled mass spectrometry facility at Ifremer (Plouzané, France) and the evaluation of $\delta^{11}$B$_{CaCO_3}$ TS4 in a diverse range of marine calcifying organisms reared for 60 days in isothermal seawater (25 °C) equilibrated with an atmospheric $p$CO$_2$ of ca. 409 µatom. Average $\delta^{11}$B$_{CaCO_3}$ composition for all species evaluated in this study range from 16.27 to 35.09‰ including, in decreasing order, coralline red alga *Neogoniolithion* sp. (35.89 ± 3.71‰), temperate coral *Oculina arbuscula* (24.12 ± 0.19‰), serpulid worm *Hydroides crucigera* (19.26 ± 0.16‰), tropical urchin *Eucidaris tribuloides* (18.71 ± 0.26‰), temperate urchin *Arbacia punctulata* (16.28 ± 0.86‰), and temperate oyster *Crassostrea virginica* (16.03‰). These results are discussed in the context of each species' proposed mechanism of biocalcification and other factors that could influence skeletal and shell $\delta^{11}$B, including calcifying site pH, the proposed direct incorporation of isotopically enriched boric acid (instead of borate) into biogenic calcium carbonate, and differences in shell/skeleton polymorph mineralogy. We conclude that the large inter-species variability in $\delta^{11}$B$_{CaCO_3}$ (ca. 20) and significant discrepancies between measured $\delta^{11}$B$_{CaCO_3}$ and $\delta^{11}$B$_{CaCO_3}$ expected from established relationships between abiogenic $\delta^{11}$B$_{CaCO_3}$ and seawater pH arise primarily from fundamental differences in calcifying site pH amongst the different species. These results highlight the potential utility of $\delta^{11}$B as a proxy of calcifying site pH for a wide range of calcifying taxa and underscore the importance of using species-specific seawater-pH–$\delta^{11}$B$_{CaCO_3}$ calibrations when reconstructing seawater pH from $\delta^{11}$B of biogenic carbonates.

## 1 Introduction

The ability to monitor historical changes in seawater pH on both short and long timescales is necessary to understand the influence that changes in the partial pressure of atmospheric CO$_2$ ($p$CO$_2$) have had on the carbonate chemistry of seawater. The recent anthropogenic increase in $p$CO$_2$ has already

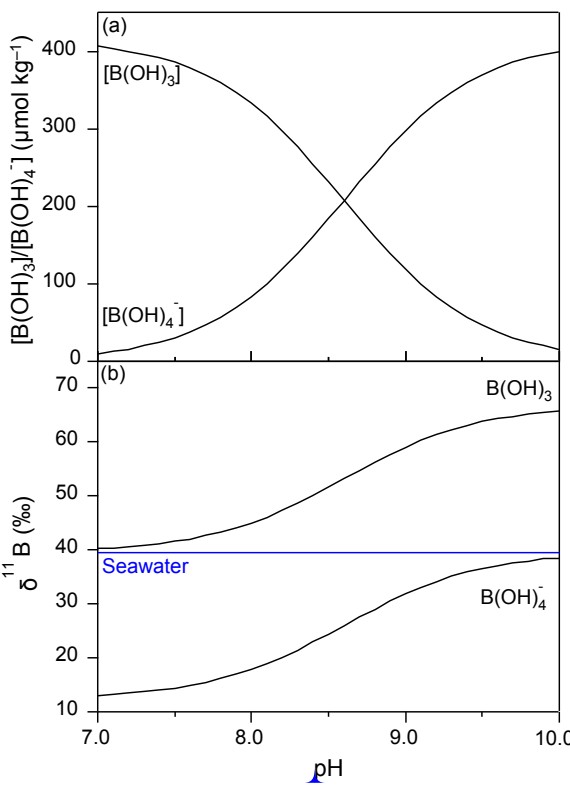

**Figure 1. (a)** Speciation of dissolved inorganic boron, $B(OH)_3$ and $B(OH)_4^-$, as a function of seawater pH. **(b)** $\delta^{11}$B of dissolved inorganic boron species as a function of seawater pH. The $pK_B$ is 8.6 at 25 °C and 35 psu (practical salinity units; Dickon, 1990), $\alpha$ is 1.0272 (Klochko et al., 2006), and $\delta^{11}B_{SW}$ is 39.61 (Foster et al., 2010).

resulted in a significant decrease in seawater pH (Bates, 2007; Byrne et al., 2010; Dore et al., 2009; Feely et al., 2008, 2016; Gonzalez-Davlia et al., 2010; IPCC, 2014), which affects the ability of marine calcifying organisms to pro-
5 duce their shells and skeletons ($CaCO_3$; IPCC, 2014). Experimental studies have revealed that organismal responses to ocean acidification vary widely amongst taxa, highlighting the complexity of biological responses to global change stressors (e.g., Kroeker et al., 2010, 2013; Ries et al., 2009)
and necessitating a more thorough understanding of how an organism's mechanism of biocalcification governs its specific response to ocean acidification.

## 1.1 Theoretical model of $\delta^{11}$B variation with pH

The boron isotope composition ($\delta^{11}$B) of biogenic $CaCO_3$
($\delta^{11}B_{CaCO_3}$) has been primarily used as a paleoceanographic proxy for seawater pH (Hönisch and Hemming, 2004; Hönisch et al., 2004; Montagna et al., 2007; Palmer, 1998; Pearson et al., 2009; Penman and Hönisch, 2014; Rae et al., 2011; Trotter et al., 2011; Vengosh et al., 1991; Wei et al.,
2009). Boron has a residence time in seawater of ca. 14 mil-

lion years (Lemarchand et al., 2000), which is much longer than the mixing time of oceans (ca. 1000 years), suggesting that it is conservatively distributed throughout the ocean (Foster et al., 2010) – making $\delta^{11}$B a potentially attractive proxy for paleo-seawater pH.

Boron exists in aqueous solutions as either trigonal boric acid [$B(OH)_3$] or as the tetrahedral borate anion [$B(OH)_4^-$], and their proportions in solution vary as a function of pH (Fig. 1) pursuant to the following equilibrium reaction:

$$B(OH)_3 + H_2O \leftrightarrow B(OH)_4^- + H^+.$$

In modern seawater, $B(OH)_4^-$ represents ca. 24.15 % of dissolved boron, assuming that the dissociation constant ($pK_B$) between the two species of boron is 8.597 (at 25 °C, pH = 8.1, 35 psu (practical salinity units; Dickson, 1990). Boron has two stable isotopes, $^{10}$B and $^{11}$B, with relative
abundances of 19.9 and 80.1 %, respectively. $B(OH)_3$ is enriched in $^{11}$B relative to $B(OH)_4^-$ due to differences in the ground state energy of molecular vibration of these chemical species in solution. The isotopic composition of boron is expressed following standard convention:

$$\delta^{11}B = \left[ \left( ^{11}B_{sample}/^{10}B_{sample} \right) \right.$$
$$\left. / \left( ^{11}B_{standard}/^{10}B_{standard} \right) - 1 \right]$$
$$\times 1000(\text{‰}),\tag{1}$$

where the reference standard is NIST SRM (Standard Reference Materials CE2) 951 (Catanzaro et al., 1970).

The $\delta^{11}$B of modern seawater is 39.61 ± 0.20‰ (Foster et al., 2010) and a large (> 20‰) and constant isotope fractionation exists between the two aqueous species described above. The fractionation factor ($\alpha$) for boric acid and borate ion is defined as

$$\alpha \equiv \frac{\left( ^{11}B/^{10}B \right)_{\text{Boric acid}}}{\left( ^{11}B/^{10}B \right)_{\text{Borate ion}}}.$$

TS5 A range of theoretical and empirical values for $\alpha$ has been suggested (Byrne et al., 2006; Kakihana et al., 1977; Klochko et al., 2006; Nir et al., 2015; Palmer et al., 1987). For example, $\alpha$ of 1.0194 was calculated from theory by
55 Kakihana et al. (1977) and was widely applied in reconstructions of paleo-seawater pH (Hönisch et al., 2004; Kakihana et al., 1977; Sanyal et al., 1995). Zeebe (2005) used analytical techniques and ab initio molecular orbital theory to calculate $\alpha$ ranging from 1.020 to 1.050 at 300 K. Zeebe (2005)
provided several arguments in support of $\alpha \geq 1.030$, ultimately concluding that experimental work was required to determine the $\alpha$ for dissolved boric acid and the borate ion. Subsequent to the work by Zeebe (2005), significant error was identified for the borate vibrational spectrum term
used in Kakihana et al.'s (1977) theoretical calculation of $\alpha$ (Klochko et al., 2006; Rustad and Bylaska, 2007). An

empirical $\alpha$ of 1.0272 (Klochko et al., 2006), using a corrected borate vibrational spectrum term, is now considered to best describe the boron isotope fractionation between dissolved boric acid and borate ion in seawater (Rollion-Bard and Erez, 2010; Xiao et al., 2014). Moreover, due to the ability of some calcifying organisms to alter carbonate chemistry at their site of calcification, paleo-seawater pH ~~may not simply be reconstructed~~ by projecting measured $\delta^{11}$B of calcium carbonate ($\delta^{11}$B$_{CaCO_3}$) onto a theoretical seawater borate $\delta^{11}$B ($\delta^{11}$B$_{B(OH)_4^-}$)–pH curve [TS6] (see also Anagnostou et al., 2012; Honïsch et al., 2003; Sanyal et al., 1996, 2001; Trotter et al., 2011). Instead, the species used for paleo-seawater pH reconstructions may require calibration through controlled laboratory experiments and/or core-top calibrations that empirically define the species-specific relationship between seawater pH (pH$_{SW}$) and $\delta^{11}$B$_{CaCO_3}$.

The $\delta^{11}$B-based paleo-seawater pH proxy is based on a theoretical model of $\delta^{11}$B$_{B(OH)_4^-}$ variation with pH described by the following equation (Zeebe and Wolf-Gladrow, 2001):

$$\text{pH} = \text{p}K_B = -\log\left(\frac{\delta^{11}B_{CaCO_3} - \delta^{11}B_{SW}}{\left(\delta^{11}B_{SW} - \alpha_B\right) \times \left(\delta^{11}B_{CaCO_3} - \varepsilon_B\right)}\right).$$

This theoretical model of $^{11}$B$_{B(OH)_4^-}$ variation as a function of seawater pH requires knowledge of the $\alpha$ for isotope exchange between the aqueous species of boron, the dissociation constant (p$K_B$), and the isotopic composition of total boron in seawater (Pagani et al., 2005) – each of which can introduce uncertainty into the pH reconstruction.

Application of this proxy also assumes that $\delta^{11}$B$_{CaCO_3}$ reflects seawater $\delta^{11}$B$_{B(OH)_4^-}$ and, thus, seawater pH (Hemming and Hanson, 1992). Although early studies assumed that $\delta^{11}$B$_{CaCO_3}$ was indeed equivalent to seawater $\delta^{11}$B$_{B(OH)_4^-}$ (e.g., Hemming and Hanson, 1992), Sanyal et al. (2000, 2001) observed that empirically derived $\delta^{11}$B$_{CaCO_3}$–pH curves of biogenic and abiogenic calcites were parallel but vertically offset from the theoretical $\delta^{11}$B$_{B(OH)_4^-}$–pH curve, which led them to conclude that paleo-seawater pH cannot always be directly calculated from $\delta^{11}$B$_{CaCO_3}$ using the theoretical $\delta^{11}$B$_{B(OH)_4^-}$–pH relationship (i.e., $\delta^{11}$B$_{CaCO_3}$–pH relationships must be empirically calibrated for the species hosting the paleo-pH proxy).

The $\delta^{11}$B-based paleo-seawater pH proxy also relies on the assumption that B(OH)$_4^-$ is the dominant species of dissolved inorganic boron incorporated into CaCO$_3$ minerals precipitated from seawater. It is well established that $\delta^{11}$B of dissolved B(OH)$_4^-$ is controlled by solution pH (cf. Hemming and Hönisch, 2007; see discussion above). Therefore, $\delta^{11}$B$_{CaCO_3}$ should reflect pH of the precipitating solution if B(OH)$_4^-$ is indeed the dominant species of dissolved inorganic boron incorporated into CaCO$_3$, which is consistent with numerous empirical studies (see Hemming and Hönisch, 2007, for summary).

More recently, however, alternative models of boron incorporation into CaCO$_3$ have been proposed (Balan et al., 2016; Klochko et al., 2009; Noireaux et al., 2015; Uchikawa et al., 2015). These alternative models present a potential challenge to the utility of boron isotopes in reconstructing calcifying fluid and paleo-seawater pH (Balan et al., 2016; Klochko et al., 2009; Mavromatis et al., 2015; Noireaux et al., 2015; Uchikawa et al., 2015). These studies present evidence consistent with the incorporation of boric acid, along with borate, into some carbonates (e.g., Noireaux et al., 2015; Uchikawa et al., 2015) and/or the occurrence of trigonal boron in the carbonate lattice due to transformation from borate during carbonate precipitation (e.g., Mavromatis et al., 2015). Some of these studies also suggest that calcite is more prone to boric acid incorporation than aragonite (e.g., Noireaux et al., 2015). However, these studies evaluated inorganic carbonates' precipitates from fluids of compositions that differed substantially from seawater; it is yet to be determined whether boric acid incorporation is equally as prevalent in carbonates that are precipitated from seawater. Nevertheless, we evaluate boric acid incorporation as an alternative to our hypothesis that calcifying fluid pH exerts primary control over the $\delta^{11}$B composition of most biogenic carbonates.

## 1.2 The role of calcification site pH in calcareous biomineralization and organisms' responses to ocean acidification

Many calcifying marine organisms, including scleractinian corals (Al-Horani et al., 2003; Cohen and Holcomb, 2009; Cohen and McConnaughey, 2003; Rollion-Bard et al., 2003, 2011b; Holcomb et al., 2010; Krief et al., 2010; Trotter et al., 2011; Ries, 2011a; Anagnostou et al., 2012; McCulloch et al., 2012; Wall et al., 2016), coralline red algae (Borowitzka and Larkum, 1987; McConnaughey and Whelan, 1997; Donald et al., 2017), calcareous green algae (Borowitzka and Larkum, 1987; De Beer and Larkum, 2001; McConnaughey and Falk, 1991), foraminifera (Rink et al., 1998; Zeebe and Sanyal, 2002), and crabs (Cameron, 1985) are thought to facilitate precipitation of their skeletal or shell CaCO$_3$ by elevating the pH at their site of calcification. The effect of pH on CaCO$_3$ chemistry at the site of calcification can be summarized by the following equilibrium reactions:

$$\text{H}_2\text{CO}_3 \leftrightarrow \text{HCO}_3^- + \text{H}^+$$

and

$$\text{HCO}_3^- \leftrightarrow \text{H}^+ + \text{CO}_3^{2-},$$

which are respectively governed by the following stoichiometric dissociation constants:

$$K_1^* = \left[\text{HCO}_3^-\right]\left[\text{H}^+\right] / \left[\text{H}_2\text{CO}_3\right]$$

and

$$K_2^* = \left[\text{CO}_3^{2-}\right]\left[\text{H}^+\right] / \left[\text{HCO}_3^-\right].$$

Thus, reducing [H$^+$] at the site of calcification shifts the carbonic acid system towards elevated [CO$_3^{2-}$], thereby increasing the CaCO$_3$ saturation state ($\Omega_{\text{CaCO}_3}$) following

$$\Omega_{\text{CaCO}_3} = \left[\text{Ca}^{2+}\right]\left[\text{CO}_3^{2-}\right]/K_{\text{sp}}^*,$$

where $K_{\text{sp}}^*$ is the stoichiometric solubility product of the appropriate CaCO$_3$ polymorph (e.g., calcite, aragonite) and is influenced by temperature and salinity.

The decrease in pH$_{\text{SW}}$ that will accompany the rise in anthropogenic atmospheric $p$CO$_2$ will reduce seawater
[CO$_3^{2-}$], which has been shown to inhibit biological deposition of CaCO$_3$, or even promote its dissolution (cf. Doney et al., 2009; Fabry et al., 2008; Kleypas et al., 2006; Kroeker et al. 2010; Langdon, 2002; Ries et al., 2009). However, if seawater is the source of an organism's calcifying fluid
(e.g., Gaetani and Cohen, 2006), then the concentration of dissolved inorganic carbon (DIC) in this fluid will increase as atmospheric $p$CO$_2$ increases. Organisms able to strongly regulate pH of their calcifying fluid (pH$_{\text{CF}}$), despite reduced external pH, should convert much of this increased DIC, oc-
curring primarily as HCO$_3^-$, back into the CO$_3^{2-}$ needed for calcification (Ries, 2011a, b; Ries et al., 2009). Thus, an organism's specific response to CO$_2$-induced ocean acidification should be ~~critically~~ dependent upon that organisms' ability to regulate pH at their site of calcification.

Marine calcifiers biomineralize in diverse ways, with some calcifers' mechanisms of biomineralization better understood than others. Corals are thought to accrete CaCO$_3$ directly from a discrete calcifying fluid (e.g., Al-Horani et al., 2003; Cohen and Holcomb, 2009; Cohen and Mc-
Connaughey, 2003 and references therein; Gaetani and Cohen, 2006; Ries, 2011a), with mineralization sites and crystal orientations being influenced by organic templates and/or calicoblastic cells (e.g., Cuif and Dauphin, 2005; Goldberg, 2001; Meibom et al., 2008; Tambutté et al., 2007). Mollusks
are also thought to precipitate their shells from a discrete calcifying fluid between the external epithelium of the mantle and the inner layer of the shell known as the extrapallial fluid (EPF; e.g., Crenshaw, 1972), with hemocytes and organic templates playing a potentially important role in crys-
tal nucleation (e.g., Marie et al., 2012; Mount et al., 2004; Weiner et al., 1984). Coralline red algae are also thought to precipitate primarily high-Mg calcite (HMC) extracellularly but within a chemically controlled fluid bound by adjacent cells. Echinoids, in contrast, are thought to initiate calcifica-
tion on Ca$^{2+}$-binding organic matrices within cellular vacuoles (Ameye et al., 1998).

Various mechanisms have been proposed for elevating pH$_{\text{CF}}$, including conventional H$^+$ channelling (McConnaughey and Falk, 1991), Ca$^{2+}$–H$^+$ exchanging AT-
50 Pase (Cohen and McConnaughey, 2003; McConnaughey and Falk, 1991; McConnaughey and Whelan, 1997), light-induced H$^+$ pumping (De Beer and Larkum, 2001), transcellular symporter and co-transporter H$^+$-solute shuttling (Mc-

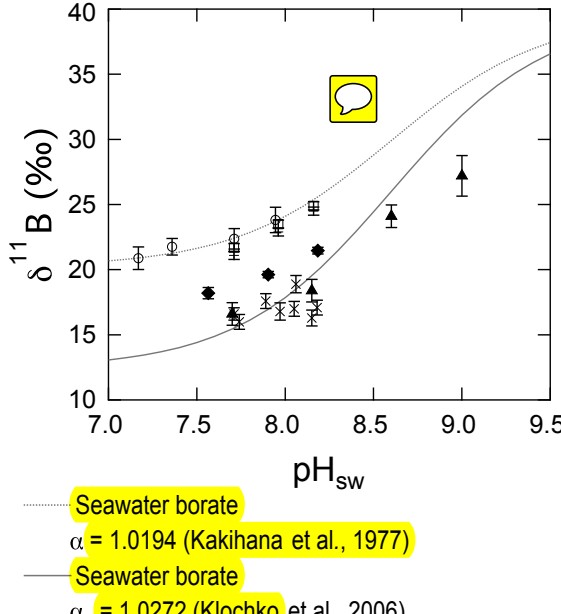

**Figure 2.** Examples of previously published $\delta^{11}$B$_{\text{CaCO}_3}$–pH$_{\text{SW}}$ trends for corals, foraminifera, and brachiopods. Although many B isotope data sets are available, only studies with $\geq 3$ $\delta^{11}$B$_{\text{CaCO}_3}$–pH$_{\text{SW}}$ data CE3 spanning a sufficiently wide range of pH$_{\text{SW}}$ conditions were selected to illustrate the range of $\delta^{11}$B$_{\text{CaCO}_3}$–pH$_{\text{SW}}$ trends published to date. The two grey lines correspond to the theoretical seawater borate $\delta^{11}$B–pH curves that have been applied most frequently to interpret $\delta^{11}$B variability in marine calcifiers. ~~The p$K_B$ is 8.6 at~~ 25 °C and 35 psu (Dickson, 1990).

Connaughey and Whelan, 1997), cellular extrusion of hydroxyl ions (OH$^-$) into the calcifying medium, and CO$_2$ con- 55 sumption via photosynthesis (e.g., Borowitzka and Larkum, 1976).

Regardless of the exact composition (e.g., seawater vs. modified seawater) or nature (e.g., fluid vs. gel) of their calcifying media, or the specific mechanisms by which they pro- 60 duce their CaCO$_3$ (e.g., organic templates vs. cellular mediation vs. proton pumps vs. Ca$^{2+}$ ATPase), an organism's ability to control pH$_{\text{CF}}$ should strongly influence their ability to convert DIC into CO$_3^{2-}$, thereby impacting their specific calcification response to CO$_2$-induced ocean acidification. 65

Please note the remarks at the end of the manuscript

### 1.3 Relationship between calcification site pH and $\delta^{11}B_{CaCO_3}$

Numerous studies have documented systematic relationships between $pH_{SW}$ and the $\delta^{11}B_{CaCO_3}$ composition of foraminiferal shells and coral skeletons (Fig. 2) that are generally consistent with theoretically derived relationships between seawater pH and $\delta^{11}B_{B(OH)_4^-}$. However, the observed relationships between biogenic $\delta^{11}B_{CaCO_3}$ and $pH_{SW}$ vary widely amongst taxa (Fig. 2) and are generally offset from that measured or derived theoretically for $B(OH)_4^-$ in seawater (Byrne et al., 2006; Klochko et al., 2006; Liu and Tossell, 2005; Zeebe, 2005) and from that observed in abiotically precipitated $CaCO_3$ (Noireaux et al., 2015; Sanyal et al., 2000).

One hypothesis for the discrepancies between the expected $\delta^{11}B_{CaCO_3}$–pH relationship and those actually observed for biogenically precipitated $CaCO_3$ is that most marine calcifiers are not precipitating their $CaCO_3$ directly from seawater, but rather from a discrete calcifying fluid with a pH ($pH_{CF}$) that is substantially elevated relative to that of their external seawater ($pH_{SW}$). For example, prior studies have shown that for a given $pH_{SW}$, $\delta^{11}B_{CaCO_3}$ of the coral species *Porites cylindrica* and *Acropora nobilis* are moderately elevated relative to $\delta^{11}B_{CaCO_3}$ of the foraminifera *Globigerinoides sacculifer* and substantially elevated relative to the mussel *Mytilus edulis* (Fig. 2; Heinemann et al., 2012; Hönisch et al., 2004; Sanyal et al., 2001). One possible explanation for these differences is that corals are maintaining their calcifying fluids at higher pH than the calcifying fluids of foraminifera, which are in turn elevated relative to the $pH_{CF}$ of mussels. This is consistent with pH microelectrode (Al Horani et al., 2003; Ries, 2011a) and fluorescent pH dye data (Venn et al., 2009, 2011, 2013), suggesting that scleractinian corals elevate their $pH_{CF}$ to 8.5–10, versus their external $pH_{SW}$ of 8; that foraminifera maintain their $pH_{CF}$ between 8 and 9 (Jorgensen et al., 1985; Rink et al., 1998); and that bivalves maintain their $pH_{CF}$ between 7.5 and 8 (Crenshaw, 1972).

Here, we investigate differences in $\delta^{11}B_{CaCO_3}$–pH relationships amongst taxonomically diverse biogenic calcification systems and discuss the compatibility of these observations with the hypothesis that $\delta^{11}B_{CaCO_3}$ of biogenic carbonate is recording $pH_{CF}$, rather than $pH_{SW}$ – a key parameter of biological calcification that has proven challenging to measure yet is fundamental to understanding, and even predicting, marine calcifiers' responses to $CO_2$-induced ocean acidification. By systematically investigating the $\delta^{11}B_{CaCO_3}$ composition of a taxonomically broad range of taxa, each employing different mechanisms of calcification yet all cultured under equivalent laboratory conditions (Ries et al., 2009), we are able to empirically assess biological controls on the $\delta^{11}B_{CaCO_3}$ composition of biogenic carbonates.

### 2 Methods and materials

#### 2.1 Laboratory conditions

Sample processing and chemical separation were performed under ISO 5 (class 100) laminar flow hoods within an ISO 6 (class 1000) clean room at Ifremer (Plouzané, France). Analyses of $^{11}B\,/\,^{10}B$ ratios were carried out using a Thermo Scientific Neptune MC-ICPMS (multi-collector inductively-coupled plasma mass spectrometer CE4) at the Pôle Spectrometrie Océan (PSO), Ifremer (Plouzané, France). Efforts were made to minimize sample exposure to laboratory air by, for example, removing caps of sample vials only when reagents were added to the samples and just prior to sample analysis.

#### 2.2 Reagents

Ultra-pure reagents were used for all chemical procedures. The source of high-purity water (UHQ) for the procedures was a Millipore Direct-Q water purification system with a specific resistivity of 18.2 M$\Omega$·cm CE5. All $HNO_3$ solutions are obtained from dilutions using Aristar ultra-high-purity acid. The 0.5 N $NH_4OH$ solutions are boron cleaned by exchange with boron-specific resin (Amberlite IRA 743). UHQ water is buffered to pH 7 with the boron-cleaned $NH_4OH$. The reagent boron blanks were measured on a Thermo Scientific Element XR at the PSO, Ifremer (Plouzané, France), and were all < 0.1 ppb, yielding a total B blank of < 100 ng per sample.

#### 2.3 Materials

##### 2.3.1 Samples

This study evaluated the $\delta^{11}B_{CaCO_3}$ of six divergent species of marine calcifiers reared for 60 days in isothermal (25 °C) and isosaline (32 psu) seawater equilibrated with atmospheric $pCO_2$ of ca. 409 μatom, including a temperate coral (*Oculina arbuscula*), a tropical coralline red alga (*Neogoniolithion* sp.), a tropical urchin (*Eucidaris tribuloides*), a temperate urchin (*Arbacia punctulata*), a serpulid worm (*Hydroides crucigera*), and a temperate oyster (*Crassostrea virginica*; see Ries et al., 2009, for details). The specimens were subsampled for new growth relative to a barium marker emplaced at the start of the experiment (details in Ries, 2011), homogenized, and at least three specimens per species analyzed for $\delta^{11}B_{CaCO_3}$.

##### 2.3.2 Standards

A range of standards were used in this study, including (1) the reference standard NIST SRM 951 (Catanzaro et al, 1970) for $\delta^{11}$B and B concentration, (2) a mixture of NIST SRM 951 and a series of ICPMS SRM for the B : Ca ratio (30–200 μg mg$^{-1}$), (3) the international coral standard (*Porites*

sp.) JCp-1 (Geological Survey of Japan, Tsukuba, Japan), (4) the international giant clam standard (*Tridacna gigas*) JCt-1 (Geological Survey of Japan, Tsukuba, Japan), and (5) a laboratory coral standard (NEP; *Porites* sp.) from the

5 University of Western Australia and the Australian National University (McCulloch et al., 2014).

## 2.4 Boron extraction procedure

Prior to boron isotope analysis, B was separated from the sample matrix using a B-specific anionic exchange resin

(Amberlite IRA-743; Kiss, 1988). Amberlite IRA 743 functions as an anion exchanger with a high affinity for B absorption at neutral to alkaline pH (i.e., will absorb B) and a low affinity for boron at acidic pH (i.e., will release B). The resin was crushed and sieved to a desired 100–200 mesh, then

cleaned and conditioned to a pH of 7 (6.8–7.2).

Two methods of B extraction are presented: batch and column chemistry. For both, the influence of matrix chemistry is removed through minor adjustments to the chemistry of existing B extraction techniques. These two methods were

20 applied to four biogenic CaCO$_3$ samples (*Porites* coral, temperate urchin, giant clam, American oyster).

### 2.4.1 Oxidative cleaning

Samples and reference materials JCp-1, JCt-1, and NEP were cleaned with an oxidative cleaning method following the

25 method of Barker et al. (2003). For a 2 mg sample, 200 µL of the alkaline-buffered (0.1 M NH$_4$OH) H$_2$O$_2$ was added to remove organic matter. Samples were placed in an ultrasonicator for 20 min at 50 °C to expedite cleaning. Following peroxide cleaning, samples were ~~then~~ submitted to multiple

washes (typically three) of UHQ water (pH = 7, 400 µL) until the pH of the supernatant matched that of the UHQ water ~~to ensure~~ removal of all the oxidizing agent. The water was then removed from samples after centrifugation and a weak-acid leach was implemented by adding 20 µL of 0.001 M HNO$_3$

to each sample. Samples were then ultrasonicated for 10 min, centrifuged, and then ~~the acid was removed~~. The samples were washed twice with pH-buffered UHQ water (buffered to pH 7 with 2 % NH$_4$OH), centrifuged, and ~~the water was removed~~. Dissolution of each sample was then performed

by addition of 20 µL of 3 M HNO$_3$ followed by 300 µL of 0.05 M HNO$_3$. The pH of each sample was then adjusted to pH 7 with 0.2 M NH$_4$OH, following ~~partition~~ coefficients for the B-specific resin reported by Lemarchand et al. (2002). ~~For both the batch and the column chemistry methods, the~~

~~resin is pre-cleaned and conditioned to pH 7 prior to sample loading.~~

### 2.4.2 Column chemistry method

A column chemistry protocol for B extraction (described in Table 1) was developed based on methods described by Wang

et al. (2010) and Foster et al. (2013). Briefly, the columns

**Table 1.** Protocol used to evaluate the column chemistry method of boron extraction. Three volumes of resin (60, 250, and 500 µL) were evaluated.

| Step | mg resin | 15 | 62.5 | 125 |
|------|----------|-----|------|-----|
| 1 | Resin (µL ) | 60 | 250 | 500 |
| 2 | UHQ H$_2$O at pH 7 (mL) | 5 | 5 | 5 |
| 3 | 0.5 N HNO$_3$ (mL) | 2.5 | 2.5 | 5 |
| 4 | UHQ H$_2$O at pH 7 (mL) ×3 | 2.5 | 2.5 | 5 |
| 5 | Check pH | | | |
| 6 | Sample Load (ng) | 536 | 536 | 536 |
| 7 | UHQ H$_2$O at pH 7 (mL) ×3 | 1 | 1 | 2 |
| 10 | Check pH | | | |
| 11 | 0.05 N HNO$_3$ (mL) | 0.5 | 0.5 | 0.5 |
| 22 | UHQ H$_2$O at pH 7 (mL) | 2 | 2 | 2 |

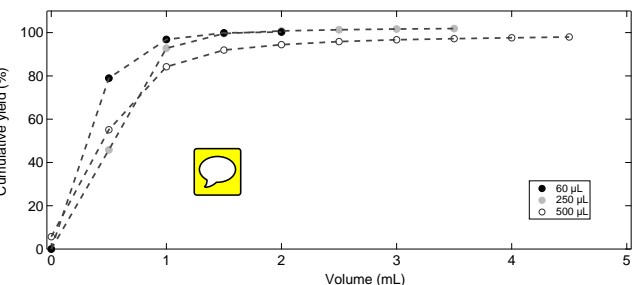

**Figure 3.** Elution curves indicating cumulative yield of boron for different volumes of the boron-specific resin (Amberlite IRA 743) used in the ion exchange column.

were washed with pH-buffered UHQ H$_2$O (pH = 7), 0.5 M HNO$_3$, and again with pH-buffered UHQ H$_2$O. After confirming that the eluent was at pH 7, the sample was loaded onto the resin and washed multiple times (1500 µL × 3) with pH-buffered UHQ in order to remove any cations, after 55 which the B was eluted in 1000 µL of 0.5 M HNO$_3$. Column yields were greater than 95 % (Fig. 3) and elution tails of every sample were checked with an additional 500 µL acid rinse. In all cases, this tail represented less than 1 % of B loaded. Small aliquots of each sample were measured 60 by single-collector HR-ICPMS (high-resolution ICPMS CE6) prior to analysis by MC-ICPMS to verify retention of B on the column and removal of other elements (e.g., Ca, Na, Ba, U).

### 2.4.3 Batch method    65

The batch method approach to B separation was conducted under closed conditions in order to reduce airborne B contamination. Cleaned samples (pH 7) were transferred into acid-cleaned microcentrifuge tubes (500 µL; polypropylene) containing 5 mg of resin (see Sect. 2.4), which is B cleaned 70 in individual tubes with 500 µL of 0.5 M HNO$_3$, and then rinsed three times with 500 µL of UHQ water to elute the other cations in the matrix and achieve pH 7. Tubes were

then capped and shaken for 15 min to promote exchange of anions from the aqueous sample to the resin. Afterwards, the mixture was centrifuged (1 min, 2000 rpm), the matrix was decanted, and the resin was washed three times (200 µL) with pH-buffered (pH 7) UHQ water to elute any cations. Boron recovery was then performed with the addition of 500 µL 0.05 M $HNO_3$ and shaken again for 15 min to promote anion exchange between the resin and solution. A final tail check was performed with 100 µL of 0.05 M $HNO_3$ to ensure that all of the B was recovered in the initial 500 µL 0.05 M $HNO_3$ solution.

## 2.5 Procedural blanks

The total yield of B from procedural blanks, which should reflect reagent, airborne and procedural contamination, was sub-nanogram (lowest yields for column and batch methods were 0.5 ng and 90 pg, respectively). Such low contamination was achieved through stringent cleaning and handling protocols for all consumables and reagents, thereby permitting accurate measurement of B at sub-µM concentrations.

## 2.6 Boron recovery and matrix removal

A major challenge in the measurement of $\delta^{11}$B by MC-ICPMS is the elimination of residual boron from prior analyses (i.e., memory effects). In order to evaluate memory effects, multiple concentrations (30 to 130 ppb) of a standard solution (NIST SRM 951) were analyzed. After washing out the MC-ICPMS with a solution of 0.05 M $HNO_3$ for several minutes, the residual $^{11}$B and $^{10}$B signals were in the range of 10–80 mV, equivalent to 5 % (30 ppb) and 3 % (130 ppb), respectively (see Fig. S1 in the Supplement for $^{11}$B blanks). Boron recovery was measured using a Thermo Scientific Element XR HR-ICP-MS at the Laboratory for Geochemistry and Metallogeny, Ifremer (Plouzané, France). Boron yields are evaluated by tracking B throughout the entire procedure.

## 2.7 Mass spectrometry

Isotopic measurements were conducted using a Thermo Scientific Neptune MC-ICPMS at the PSO, Ifremer (Plouzané, France), operated with standard plasma settings. To account for drift in mass discrimination through the analysis, samples were bracketed by matrix-matched standards of similar composition. Typically, the concentration of the standard (NIST SRM 951) was 50 ppb in 0.05 M $HNO_3$. Each analysis consisted of a 2 min simultaneous collection of masses 11 and 10 on Faraday cups H3 and L3 equipped with $10^{11}\,\Omega$ resistors. Each sample was analyzed in duplicate during a single analytical session, with replicate analyses not sharing a bracketing standard. The boron isotope ratios are reported as delta values ($\delta^{11}$B). The $\delta^{11}$B of the calcium carbonate standards JCp-1 (*Porites* sp.), NEP (*Porites* sp.), and JCt-1 (hard clam) standards, which were processed in the same manner and are reported in the results section (see Sect. 3.1.1) along-

side their published reference values (Foster et al., 2013; McCulloch et al., 2014) CE7.

The MC-ICPMS is commonly used to measure $\delta^{11}$B due to its capacity for rapid, accurate, and reproducible analyses (see McCulloch et al., 2014, for a recent summary of these methods). Challenges with this method arise from the volatile and persistent nature of boron that can result in significant memory effects, cross-contamination between samples and standards, and unanticipated matrix effects (McCulloch et al., 2014; Foster et al., 2013). Given the sensitivity of $\delta^{11}$B$_{CaCO_3}$-based estimates of pH$_{CF}$ to the analytical uncertainty cited above, two different injection methods (described below) were evaluated to determine which is most suitable for minimizing analytical error.

### 2.7.1 Demountable direct injection nebulizer

Memory effects, as described above in Sect. 2.7, were addressed by introducing samples to the plasma with a demountable direct injection high-efficiency nebulizer (*d*-DIHEN; Louvat et al., 2014). The *d*-DIHEN method minimizes the influence of memory effects by eliminating the use of a spray chamber and directly injecting the sample into the plasma (see Louvat et al., 2014, for details). Baseline B concentrations between samples were measured with counting times of 30 s (Table 2).

### 2.7.2 Ammonia addition

For the ammonia-addition method, a dual-inlet PFA Teflon spray chamber was used with an ESI PFA 50 µL min$^{-1}$ nebulizer to add ammonia gas at a rate of ca. 3 mL min$^{-1}$ (Al-Ammar et al., 2000; Foster, 2008). The addition of ammonia gas to the spray chamber ensures that the analyte remains alkaline, which prevents volatile boron from recondensing in the chamber during analysis (Al-Ammar et al., 2000). The measured B isotope signal of the rinse blank was then subtracted from the B isotope ratios in order to monitor B washout, as suggested by Foster (2008). In all cases, washout time was 200 s and samples were matrix and intensity matched to the bracketing standards.

## 3 Results

### 3.1 Method development

The yields for boron extraction for both methods were evaluated for various biogenic $CaCO_3$ samples and were typically between 97 and 102 % (determined by HR-ICPMS; see Sect. 2.6). Washes with pH-buffered UHQ $H_2O$ effectively removed Ca (99.9 %), Na (100 %), Ba (> 80 %), and U (> 93 %) from the sample matrix. The robustness of the methods is demonstrated by the observed agreement (represented as 2 standard deviations around the mean; 2SD) between measured values of the international $CaCO_3$ standards JCp-1

**Table 2.** Mass spectrometer operating conditions.

|  | $d$-DIHEN | Ammonia addition |
|---|---|---|
| Injection system | Demountable direct injection High-efficiency nebulizer | PFA Teflon spray chamber with ESI PFA Teflon $50\,\mu$L min$^{-1}$ nebulizer |
| Sample gas flow rate | $0.3$ L min$^-$ | $1.1$ L min$^{-1}$ |
| Running concentrations | B $= 50$ ppb | B $= 30$–$50$ ppb (evaluated 30, 65, 130 ppb) |
| Sensitivity | $35$ V ppm$^{-1}$, total B | $20$ V ppm$^{-1}$, total B |
| Blank level | $< 0.5$ % of $^{11}$B signal after 30s in 2 % HNO$_3$, 0.1 % after 120 s | $< 5$ % of $^{11}$B signal after 30 s in 0.05 % HNO$_3$, 3 % after 120 s |
| Resolution | Low | Low |
| Forward power | $1200$ W | $1200$ W |
| Accelerating voltage | $10$ kV | $10$ kV |
| Plasma mode | Wet plasma | Wet plasma |
| Cool gas flow rate | $16$ L min$^{-1}$ | $16$ L min$^{-1}$ |
| Auxiliary gas flow rate | $0.9$ L min$^{-1}$ | $0.9$ L min$^{-1}$ |
| Sampler cone | Standard Ni cone | Standard Ni cone |
| Skimmer cone | X Ni cone | X Ni cone |
| Interferences | $^{40}$Ar$^{++++}$ $^{20}$Ne$^{++}$ resolved | $^{40}$Ar$^{++++}$ $^{20}$Ne$^{++}$ resolved |
| Accuracy | $0.2$‰, 2sd, $n = 6$ | $0.2$‰, 2sd, $n = 6$ |
| Acquisition | $30 \times 4$ s | $30 \times 4$ s |
| Baselines | Counting times of 20 s | Counting times of 20 s |

and JCt-1, a coral (*Porites* sp.; $\delta^{11}$B$_{\mathrm{NH_3}} = 24.45 \pm 0.28$‰, $\delta^{11}$B$_{d-\mathrm{DIHEN}} = 24.30 \pm 0.16$‰) and a giant clam (*Tridacna gigas*; $\delta^{11}$B$_{\mathrm{NH_3}} = 16.65 \pm 0.39$‰, $\delta^{11}$B$_{d-\mathrm{DIHEN}} = 17.5 \pm 0.69$‰), and their values established via inter-laboratory calibration ($\delta^{11}$B $= 24.36 \pm 0.51$‰, $n = 10$ and $16.34 \pm 0.64$‰, respectively; Gutjahr et al., 2014; see Table 3). In addition, both column and batch methods were evaluated using the NEP laboratory standard (*Porites* sp.), a temperate urchin, a hard clam, and an oyster. As shown in Table 3, good agreement was achieved between $\delta^{11}$B$_{\mathrm{CaCO_3}}$ obtained via the batch and column chemistry methods for each of the biogenic CaCO$_3$ samples analyzed.

## 3.2  Boron isotope composition of marine biogenic CaCO$_3$

Average $\delta^{11}$B$_{\mathrm{CaCO_3}}$ composition for all species evaluated in this study range from 16.27 to 35.09 ‰ (Table 3). The individual and average data are presented in Tables 3 and 4, respectively, and summarized in the text that follows. Note that the variance of the data presented in Table 4 represents inter-specimen variability (i.e., variability amongst different specimens of the same species), which is substantially greater than the intra-specimen variability (i.e., variability within a specimen) and analytical variability (variability amongst repeat analyses of the same subsample of a specimen; Table 3). The coralline red alga *Neogoniolithion* sp. ($35.89 \pm 3.71$‰; $n = 3$) exhibited the highest $\delta^{11}$B$_{\mathrm{CaCO_3}}$, followed by the temperate coral *O. arbuscula* ($24.12 \pm 0.19$‰; $n = 3$), the tube of the serpulid worm *H. crucigera* ($19.26 \pm 0.16$‰; $n = 3$), the tropical urchin *E. tribuloides* ($18.71 \pm 0.26$‰; $n = 3$),

the temperate urchin *A. punctulata* ($16.28 \pm 0.86$‰; $n = 3$), and the temperate oyster *C. virginica* ($16.03$‰; $n = 1$). Therefore, a range of ca. 20‰ in $\delta^{11}$B$_{\mathrm{CaCO_3}}$ was observed across all species evaluated in this study (Tables 3 and 4). Notably, these are the first published $\delta^{11}$B$_{\mathrm{CaCO_3}}$ data for serpulid worm tubes and oysters.

## 3.3  Compatibility of the interspecific range of $\delta^{11}$B$_{\mathrm{CaCO_3}}$ with established seawater borate $\delta^{11}$B–pH relationships

Because the investigated species were cultured under relatively equivalent conditions ($p$CO$_2$ of $409 \pm 6$ µatom, $32 \pm 0.2$ psu, $25 \pm 0.1$ °C; see Ries et al., 2009), differences in pH$_{\mathrm{SW}}$ could not have been a significant driver of the observed interspecific variability in $\delta^{11}$B$_{\mathrm{CaCO_3}}$ (ca. 20‰; Tables 3 and 4). In order to evaluate this ca. 20‰ interspecific variability in $\delta^{11}$B, the data are plotted against measured pH$_{\mathrm{SW}}$ and graphically compared with theoretical borate $\delta^{11}$B–pH curves often used to interpret $\delta^{11}$B$_{\mathrm{CaCO_3}}$ data in the context of pH$_{\mathrm{SW}}$ (Fig. 4). Clear offsets from the seawater borate $\delta^{11}$B–pH curve (Klochko et al., 2006) can be observed for several of the species: the temperate coral (*O. arbuscula*) and coralline red alga (*Neogoniolithion* sp.) fall above the curve, the temperate urchin (*A. punctulata*) and American oyster (*C. virginica*) fall below the curve, and the tube of the serpulid worm (*H. crucigera*) and the tropical urchin (*E. tribuloides*) fall nearly on the curve (see Fig. 4 and Table 3). The interpretation of these offsets from the seawater borate $\delta^{11}$B–pH curve is discussed below.

**Table 3.** Boron isotope composition ($\delta^{11}$B; ‰) of all species evaluated, including international carbonate standards JCp-1 (coral, *Porites* sp.) and JCt-1 (giant clam, *Tridacna gigas*). Data are presented as the average of $n$ analyses and the precision is reported as 2 standard deviations (2SD). The cleaning protocol (oxidized, Ox; uncleaned, U), separation method (column, batch), and injection method (NH$_3$, $d$-DIHEN) are presented for comparison.

| Sample type | Name | $\delta^{11}$B | (2SD) | $n$ | Cleaning | Separation | Injection |
|---|---|---|---|---|---|---|---|
| Giant clam | JCt-1 | 17.50 | 0.69 | 6 | Ox | batch | $d$-DIHEN |
| Giant clam | JCt-1 | 16.90 | 0.30 | 6 | Ox | batch | NH$_3$ |
| Giant clam | JCt-1 | 16.34 | 0.64 | 2 | Ox | column | NH$_3$ |
| Giant clam | JCt-1 | 16.24 | 0.42 | 2 | U | batch | NH$_3$ |
| *Porites* coral | JCp-1 | 24.52 | 0.34 | 6 | Ox | column | NH$_3$ |
| *Porites* coral | JCp-1 | 24.30 | 0.16 | 10 | Ox | batch | $d$-DIHEN |
| *Porites* coral | JCp-1 | 24.65 | 0.60 | 6 | Ox | batch | NH$_3$ |
| *Porites* coral | JCp-1 | 24.44 | 0.56 | 6 | U | column | NH$_3$ |
| *Porites* coral | JCp-1 | 24.41 | 0.30 | 6 | U | batch | NH$_3$ |
| *Porites* coral | JCp-1 | 24.36 | 0.51 | 2 | Ox | column | NH$_3$ |
| *Porites* coral | JCp-1 | 24.24 | 0.38 | 2 | Ox | batch | NH$_3$ |
| *Porites* coral | NEP-1 | 26.56 | 0.34 | 2 | U | batch | NH$_3$ |
| *Porites* coral | NEP-1 | 25.51 | 0.38 | 2 | Ox | column | NH$_3$ |
| *Porites* coral | NEP-1 | 25.34 | 0.78 | 2 | Ox | batch | NH$_3$ |
| *Porites* coral | NEP-1 | 25.52 | 0.46 | 2 | U | column | NH$_3$ |
| *Porites* coral | NEP-1 | 25.92 | 0.12 | 2 | U | batch | NH$_3$ |
| *Porites* coral | NEP-1 | 25.96 | 0.30 | 2 | Ox | batch | NH$_3$ |
| Temperate coral | OCU-9 | 24.04 | na | 1 | Ox | batch | NH$_3$ |
| Temperate coral | OCU-10 | 23.98 | na | 1 | Ox | batch | NH$_3$ |
| Temperate coral | OCU-11 | 24.34 | na | 1 | Ox | batch | NH$_3$ |
| Coralline alga | JR-19 | 39.94 | 0.12 | 2 | Ox | batch | NH$_3$ |
| Coralline alga | JR-20 | 32.65 | 0.46 | 2 | Ox | batch | NH$_3$ |
| Coralline alga | JR-20 | 32.68 | 0.22 | 2 | Ox | column | NH$_3$ |
| Coralline alga | JR-21 | 35.07 | na | 1 | Ox | batch | NH$_3$ |
| Tropical urchin | JR-56 | 19.00 | 0.36 | 2 | Ox | batch | NH$_3$ |
| Tropical urchin | JR-57 | 18.64 | 0.11 | 2 | Ox | batch | NH$_3$ |
| Tropical urchin | JR-58 | 18.49 | 0.09 | 2 | Ox | batch | NH$_3$ |
| Temperate urchin | JR-64 | 14.96 | 0.10 | 2 | Ox | column | NH$_3$ |
| Temperate urchin | JR-64 | 17.60 | 0.80 | 2 | Ox | batch | $d$-DIHEN |
| Temperate urchin | JR-65 | 17.11 | 1.10 | 2 | Ox | batch | NH$_3$ |
| Temperate urchin | JR-66 | 15.43 | 0.11 | 2 | Ox | batch | NH$_3$ |
| Serpulid worm tube | JR-1 | 19.44 | na | 1 | Ox | batch | NH$_3$ |
| Serpulid worm tube | JR-2 | 19.13 | na | 1 | Ox | batch | NH$_3$ |
| Serpulid worm tube | JR-3 | 19.21 | na | 1 | Ox | batch | NH$_3$ |
| American oyster | JR125 | 16.18 | 0.16 | 2 | Ox | column | NH$_3$ |
| American oyster | JR125 | 15.90 | 0.60 | 2 | Ox | batch | $d$-DIHEN |
| American oyster | JR125 | 16.00 | 0.32 | 2 | U | batch | NH$_3$ |

# 4 Discussion

## 4.1 Appropriateness of method for analyzing $\delta^{11}$B$_{CaCO_3}$ in marine CaCO$_3$ samples

This study describes extensive method development and analytical validation used to establish stable boron isotope measurements at Ifremer (Plouzané, France), including comparisons of different techniques for sample preparation and for sample introduction to the mass spectrometer. For each of the samples evaluated, neither cleaning protocol, nor method of sample preparation, nor injection system was found to cause a significant ($p$-value $< 0.05$) difference in the $\delta^{11}$B$_{CaCO_3}$ composition of the samples (Table 3). The most effective method for minimizing memory effects in the MC-ICPMS

**Table 4.** Summary of the average and standard deviation (SD) of $\delta^{11}$B for each species (‰), calculated pH of calcifying fluid (pH$_{CF}$), pH of seawater (pH$_{SW}$) during the experimental conditions, difference between pH$_{CF}$ and pH$_{SW}$ ($\Delta$pH), calcification response to ocean acidification experiments (OA response; Ries et al., 2009), and shell/skeletal mineralogy (HMC = high-Mg calcite; LMC = low-Mg calcite; Ries et al., 2009). In most cases three biological replicates of each species were analyzed. NA = not available CE9 , only one biological replicate analyzed.

| Sample type | Scientific name | $\delta^{11}$B (SD) | pH$_{CF}$ | pH$_{SW}$ | $\Delta$pH | OA response | Mineralogy |
|---|---|---|---|---|---|---|---|
| Coralline alga | *Neogoniolithion* sp. | 35.89 (3.71) | 9.4 | 8.1 | 1.3 | Parabolic | HMC |
| Temperate coral | *Oculina arbuscula* | 24.12 (0.19) | 8.5 | 8.1 | 0.4 | Threshold | Aragonite |
| Tropical urchin | *Eucidaris tribuloides* | 18.71 (0.26) | 8.1 | 8.0 | 0.1 | Threshold | HMC |
| Serpulid worm | *Hydroides crucigera* | 19.26 (0.16) | 8.2 | 8.1 | 0.1 | Negative | Aragonite + HMC |
| Temperate urchin | *Arbacia punctulata* | 16.28 (0.86) | 7.9 | 8.0 | −0.1 | Parabolic | HMC |
| American oyster | *Crassostrea virginica* | 16.03 (NA) | 7.9 | 8.2 | −0.3 | Negative | LMC |

Note: SD is calculated from measurements of different individuals of the same species, thereby reflecting interspecimen variability. Variability arising from intra-specimen variation (i.e., variability within a single specimen) and analytical error is ~~provided~~ in Table 3.

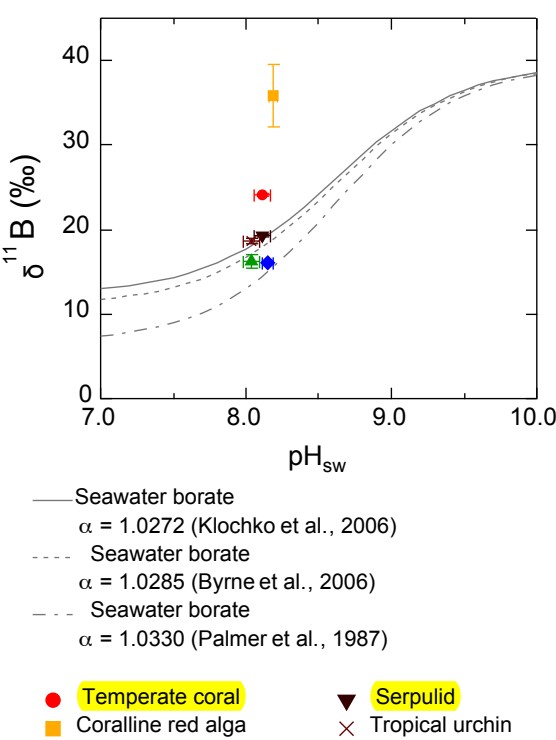

**Figure 4.** Boron isotopic composition (±SD) of different species of marine calcifiers as a function of seawater pH (±SD). The six species shown in this figure were grown under controlled $p$CO$_2$ conditions of ca. 409 ppm. Grey lines are theoretical seawater $\delta^{11}$B$_{B(OH)_4^-}$–pH curves based on different $\alpha$ that have been used to describe boron isotope fractionation between borate ion and boric acid in seawater (using pK$_B$ of 8.6152 at 25 °C and 32 psu). Although $\alpha = 1.0272$ (Klochko et al., 2006) is presently the most commonly used $\alpha$, $\delta^{11}$B$_{B(OH)_4^-}$–pH curves calculated from other values of $\alpha$ are also shown for comparison.

analyses was found to be $d$-DIHEN (Louvat et al., 2011). However, $d$-DIHEN has a complicated set-up and often generates capillary blockages arising from the aspiration of particles (e.g., resin) and/or from plasma extinction resulting from air bubble introduction. In short, sample analysis via $d$-DIHEN requires nearly continuous use to maintain its stability. In contrast, the ammonia-addition method (Al-Ammar et al., 1999, 2000) requires continuous attention by personnel while in use, due to the use of ammonia gas, but is set up and disassembled with relative ease between uses. A constant ammonia flow of 3 mL min$^{-1}$ was necessary to maintain a sufficiently high pH to enable a fast rinse. Less than a 3 % boron memory effect was stable after 2 min, enabling a signal correction for the f~~ollowing~~ sample. Both the column and batch methods of B separation yielded low blanks when < 60 µL of resin was used (see Sect. 2.5 and 2.6). However, the batch method was identified as preferable over the column chemistry method since the batch method has a lower risk of B contamination due to reduced contact time with air and the small volumes of both resin and acids (both potential sources of contamination) used in the separation process.

### 4.2 The $\delta^{11}$B$_{CaCO_3}$ compositions of a diverse range of marine calcifiers

The six species investigated exhibited a broad spectrum of $\delta^{11}$B$_{CaCO_3}$ compositions, ranging from 16.03 to 35.89‰ (Table 4) despite exposure of all species to an approximately equivalent pH$_{SW}$ of 8 (see Table 4). ~~We cannot constrain whether the relationship between $\delta^{11}$B$_{CaCO_3}$ and $\delta^{11}$B of borate significantly differs from unity in this experiment with a single pH$_{SW}$.~~ Because $\delta^{11}$B$_{B(OH)_4^-}$ at the species' sites of calcification cannot be measured or calculated from the data at hand, it cannot be directly compared with the measured $\delta^{11}$B$_{CaCO_3}$ to determine if $\delta^{11}$B$_{CaCO_3}$ necessarily reflects calcifying fluid $\delta^{11}$B$_{B(OH)_4^-}$ and, thus, pH$_{CF}$. Assuming that only the borate ion is incorporated into biogenic CaCO$_3$ (i.e., $\delta^{11}$B$_{CaCO_3}$ = calcifying fluid $\delta^{11}$B$_{B(OH)_4}$), the

wide variation in $\delta^{11}\text{B}_{\text{CaCO}_3}$ (ca. 20 ‰) amongst the investigated species reared under equivalent thermochemical conditions may indeed arise from inherent differences in pH$_{\text{CF}}$ amongst the species. If this is the case, then the observed range in $\delta^{11}\text{B}_{\text{CaCO}_3}$ amongst the species (16.03 to 35.89 ‰) translates to an approximate range in pH$_{\text{CF}}$ of 7.9–9.4.

The amount of boron (i.e., ~~B : Ca~~ co-precipitated with inorganic (i.e., abiogenic) CaCO$_3$ is known to be dependent on solution pH and inorganic CaCO$_3$ precipitation rate. However, the relative abundances of the inorganic B species in solution that are incorporated into inorganic CaCO$_3$ (borate ion and boric acid) have been shown to be independent of parent solution pH (Mavromatis et al., 2015). Although Mavromatis et al. (2015) also found that polymorph mineralogy influences both the B : Ca ratio (higher in aragonite than calcite) and coordination of B in inorganic CaCO$_3$ (tetrahedral / trigonal ratio higher in aragonite than in calcite), the B : Ca ratio alone does not appear to influence boron isotope fractionation in CaCO$_3$ (Noireaux et al., 2015). It should also be noted that these experiments (Mavromatis et al., 2015; Noireaux et al., 2015) analyzed carbonates precipitated from non-seawater solutions; therefore, further work is needed to determine the applicability of these findings to marine carbonates. Furthermore, because the borate / boric acid ratio is higher in aragonite than in calcite, aragonite-producing species (corals, serpulid worms) should have a universally lower $\delta^{11}\text{B}_{\text{CaCO}_3}$ composition than calcite-producing species (urchins, coralline algae, oysters) if shell mineralogy was the primary driver of the observed interspecific variation in $\delta^{11}\text{B}_{\text{CaCO}_3}$ compositions – a trend that is not observed (Fig. 4). Thus, interspecific differences in polymorph mineralogy cannot, alone, explain the species' disparate $\delta^{11}\text{B}_{\text{CaCO}_3}$ compositions. The more parsimonious explanation for these observed differences in $\delta^{11}\text{B}_{\text{CaCO}_3}$ appears to be differences in pH$_{\text{CF}}$, which would change the speciation of dissolved B at the site of calcification and therefore the isotopic composition of the borate ion that is preferentially incorporated into the organisms' CaCO$_3$.

Significant deviations from equilibrium exist in the stable isotopic compositions (e.g., O, C, B) of biogenic marine CaCO$_3$ (e.g., Hemming and Hanson, 1992; McConnaughey, 1989). Notably, many marine calcifiers exhibit $\delta^{11}\text{B}_{\text{CaCO}_3}$ that differs from the $\delta^{11}\text{B}_{\text{B(OH)}_4^-}$ of their surrounding seawater (Figs. 3 and 5). When interpreted in the context of the framework that skeletal $\delta^{11}$B reflects pH$_{\text{CF}}$ rather than the organism's ambient pH$_{\text{SW}}$, these results suggest that marine calcifiers are precipitating their CaCO$_2$ from a discrete fluid with a pH$_{\text{CF}}$ higher than, equal to, or, for some species, below that of seawater. A second hypothesis is that pH$_{\text{CF}}$ exerts some control over $\delta^{11}\text{B}_{\text{B(OH)}_4^-}$ at the site of calcification and, hence, $\delta^{11}\text{B}_{\text{CaCO}_3}$, but that there are other species-specific effects that also influence $\delta^{11}\text{B}_{\text{CaCO}_3}$ composition. The compatibility of these two hypotheses with existing models of biomineralization and observed $\delta^{11}\text{B}_{\text{CaCO}_3}$ for the various

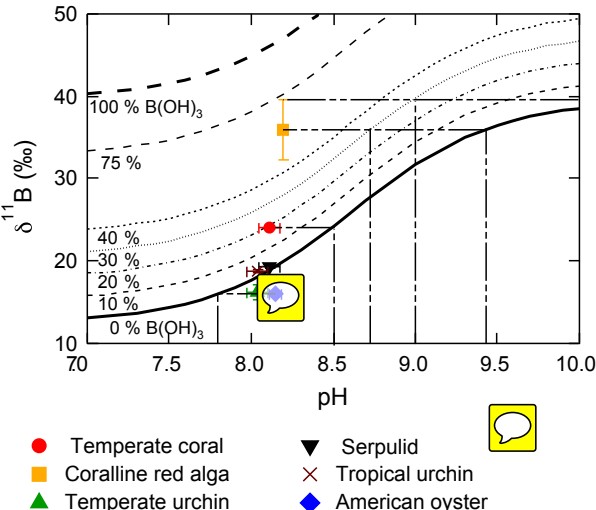

**Figure 5.** Exploring the potential influence of pH and boron speciation on $\delta^{11}\text{B}_{\text{CaCO}_3}$ (adapted from Rollion-Bard et al., 2011b). The solid and dashed curves represent the $\delta^{11}\text{B}_{\text{CaCO}_3}$ that would result from the incorporation of different amounts of B(OH)$_3$ into the biogenic carbonates. The dashed vertical lines represent the calculated pH$_{\text{CF}}$ based on the assumptions that 0 % B(OH)$_3$ is incorporated into the temperate coral skeleton and that ~~0, 30, and 75~~ % B(OH)$_3$ is incorporated into the coralline algal skeleton. Of all species examined, only the coralline algae has a $\delta^{11}\text{B}_{\text{CaCO}_3}$ composition that could conceivably originate at least in part from B(OH)$_3$ incorporation, although this would require a much higher level (ca. 3-fold) of skeletal B(OH)$_3$ incorporation than has been previously observed (e.g., Cusack et al., 2015; see text for details).

marine calcifiers investigated in the present study are discussed below.

### 4.2.1 Coralline red alga (*Neogoniolithon* sp.)

Coralline red algae ~~are also thought to~~ precipitate primarily high-Mg calcite from a calcifying fluid bounded by adjacent cells (Simkiss and Wilbur, 1989). Thus, biomineralization by coralline red algae occurs extracellularly but primarily within a chemically controlled environment within and adjacent to cell walls, with calcite crystals exhibiting preferred orientations – atypical of calcifying macroalgae (Simkiss and Wilbur, 1989). The average $\delta^{11}\text{B}_{\text{CaCO}_3}$ for the branching, non-articulated coralline red alga *Neogoniolithion* sp. evaluated in this study ($35.89 \pm 3.71$ ‰; $n = 3$; Tables 3 and 4) is higher than the $\delta^{11}\text{B}_{\text{CaCO}_3}$ composition of any other calcifying marine organism evaluated to date (Table 5). Of particular interest, one of the coralline red alga specimens evaluated in this study exhibited $\delta^{11}\text{B}_{\text{CaCO}_3}$ (39.94 ‰, Table 3) similar to the average $\delta^{11}$B of the total dissolved boron in seawater (i.e., comprising the $\delta^{11}$B composition of both dissolved borate and boric acid; 39.61 ‰) determined by Foster et al. (2010), raising the possibility that coralline red algae incorporate both species of dissolved inorganic boron

**Table 5.** Previously published $\delta^{11}$B analyses of biogenic marine carbonates and seawater samples.

| Sample | Mineralogy | $\delta^{11}$B range (‰) | Reference |
|---|---|---|---|
| Modern coral | Aragonite | 26.7–31.9 | Vengosh et al. (1991) |
| Modern coral | Aragonite | 23.0–24.7 | Hemming and Hanson (1992) |
| Modern coral | Aragonite | 23.5–27.0 | Gaillardet and Allègre (1995) |
| Modern coral | Aragonite | 23.9–26.2 | Hemming et al. (1998) |
| Modern coral | Aragonite | 25.2 | Allison and Finch (2010) |
| Modern coral | Aragonite | 23.56–27.88 | Anagnostou et al. (2012) |
| Modern coral | Aragonite | 21.5–28.0 | Dishon et al. (2015) |
| Modern coral | Aragonite | 21.76–23.19 | Dissard et al. (2012) |
| Deep sea coral | Calcitic | 13.7–17.3 | Farmer et al. (2015) |
| Modern coral | Aragonite | 18.52–23.96 | Holcomb et al. (2014) |
| Modern coral | Aragonite | 21.1–24.9 | Hönisch et al. (2004) |
| Modern coral | Aragonite | 23.2–28.7 | McCulloch et al. (2012) |
| Deep sea coral | Calcitic | 15.5 | McCulloch et al. (2012) |
| Modern coral | Aragonite | 22.5–24.0 | Reynaud et al. (2004) |
| Modern coral | Aragonite | 31.1–35.7 | Rollion-Bard et al. (2011a) |
| Modern coral | Aragonite | 18.6–30.6 | Rollion-Bard et al. (2011b) |
| Modern coral | Aragonite | 21–24.5 | Schoepf et al. (2014) |
| Modern coral | Aragonite | 23.6–25.2 | D'Olivo et al. (2015) |
| Ancient coral | Aragonite | 23.6–27.1 | Douville et al. (2010) |
| Ancient coral | Aragonite | 24.5–27.1 | Kubota et al. (2014) |
| Ancient coral | Aragonite | 22.5–25.5 | Liu et al. (2009) |
| Modern coral | Aragonite | 21.1–25.4 | Wei et al. (2009) |
| Planktonic foraminifera | Calcite | 14.2–19.8 | Vengosh et al. (1991) |
| Planktonic foraminifera | Calcite | 22.0–23.3 | Sanyal et al. (1995) |
| Planktonic foraminifera | Calcite | 18.4 | Sanyal et al. (1997) |
| Benthic foraminifera | Calcite | 13.3, 20.3, 32.0 | Vengosh et al. (1991) |
| Benthic foraminifera | Calcite | 20.5, 21.4 | Sanyal et al. (1995) |
| Bulk foraminifera | Calcite | 10.5, 11.5, 14.8, 16.2, 17.0 | Spivak et al. (1993) |
| Planktonic foraminifera | Calcite | 17.1, 22.9 | Kasemann et al. (2009) |
| Planktonic foraminifera | Calcite | 20.6–25.4 | Ni et al. (2007) |
| Benthic foraminifera | Calcite | 14.5–16.8 | Rae et al. (2011) |
| Benthic foraminifera | Calcite | 18–30.1 | Rollion-Bard and Erez (2010) |
| Benthic foraminifera | Calcite | 15.8–17.4 | Yu et al. (2010) |
| Planktonic foraminifera | Calcite | 16.9–17.9 | Yu et al. (2013) |
| Planktonic foraminifera | Calcite | 19.1–22.2 | Bartoli et al. (2011) |
| Planktonic foraminifera | Calcite | 16.2–19.8 | Foster (2008) |
| Planktonic foraminifera | Calcite | 15.2–17.2 | Foster et al. (2012) |
| Benthic foraminifera | Calcite | 13.09–13.37 | Foster et al. (2012) |
| Planktonic foraminifera | Calcite | 18.9–21.8 | Foster and Sexton (2014) |
| Planktonic foraminifera | Calcite | 20.8–23.3 | Hönisch and Hemming (2005) |
| Planktonic foraminifera | Calcite | 21.7–23.4 | Hönisch et al. (2009) |
| Benthic foraminifera | Calcite | 18.0 | Kaczmarek et al. (2015) |
| Planktonic foraminifera | Calcite | 15.1–16.4, 18.9–21.4 | Martínez-Botí et al. (2015a) |
| Planktonic foraminifera | Calcite | 19.1–19.8, 19.4–20.8 | Martínez-Botí et al. (2015b) |
| Planktonic foraminifera | Calcite | 24.2–25.7 | Palmer et al. (2010) |
| Mixed foraminifera | Calcite | 19.4–27.7 | Palmer (1998) |
| Mixed foraminifera | Calcite | 20.8–26.6 | Pearson and Palmer (1999) |
| Planktonic foraminifera | Calcite | 11–13.5∗, 21.6–25.5 | Pearson and Palmer (2000) |
| Benthic foraminifera | Calcite | 15.2–16.2 | Rae et al. (2014) |
| Planktonic foraminifera | Calcite | 13.6–15.8 | Penman and Hönisch (2014) |
| Echinoid | High-Mg calcite | 22.7–22.9 | Hemming and Hanson (1992) |
| Goniolithon | High-Mg calcite | 22.4 | Hemming and Hanson (1992) |
| Encrusting red algae | High-Mg calcite | 23.0 | Hemming and Hanson (1992) |
| Thecidellina | Calcite | 21.5–22.5 | Hemming and Hanson (1992) |
| Other carbonates | Aragonite | 19.1–24.8 | Hemming and Hanson (1992) |
| Seawater | Seawater | 39.9–40.2 | Hemming and Hanson (1992) |
| Seawater | Seawater | 37.7–40.4 | Foster et al. (2010) |

during calcification. In support of this argument, Cusack et al. (2015) provide NMR data indicating that 30 % of the B incorporated into the coralline red alga *Lithothamnion glaciale* was present as boric acid. However, since the coralline red algae were reared at a $pH_{SW}$ of 8.1, the $\delta^{11}B_{CaCO_3}$ compositions observed for the coralline algae in the present study would require incorporation of both inorganic species of boron at $[B(OH)_3] : [B(OH)_4^-]$ ratios of ca. 3 : 1, which is not consistent with prior observations for inorganic and biogenic calcite. For example, Cusack et al. (2015) reported 30 % trigonal boron in the calcite lattice of a different species of coralline alga. Therefore, boric acid incorporation alone cannot explain the anomalously elevated $\delta^{11}B_{CaCO_3}$ observed here for coralline algae (see also discussion in Donald et al., 2017). Moreover, although nuclear magnetic resonance spectroscopy reveals that trigonal boron is present in the calcite lattice, it cannot determine whether boric acid was incorporated directly into the calcite lattice, or if the trigonal boron originated from borate post-mineralization (e.g., see alternative mechanisms of boron incorporation discussed in Klochko, 2006; Noireaux et al., 2015). Nevertheless, if 30 % of the B in the calcite lattice of coralline algal skeleton is indeed incorporated directly as trigonal boron, as reported by Cusack et al. (2015), $pH_{CF}$ would still need to be as high as 9 to explain the anomalously high $\delta^{11}B_{CaCO_3}$ (see Fig. 5). Short et al. (2015) observed that epiphytic turf algae can increase $pH_{SW}$ up to 9 within their diffusive boundary layer, driven by the algae's photosynthetic drawdown of aqueous $CO_2$, lending further support to the idea that other types of algae, such as coralline red algae, could maintain their calcifying fluid at or above pH 9. Thus, $\delta^{11}B_{CaCO_3}$ compositions of coralline red algae may indeed reflect substantially elevated $pH_{CF}$ (9.4; Table 4, Fig. 4), suggesting that coralline red algae are highly efficient at removing protons and/or dissolved inorganic carbon from their calcifying medium.

### 4.2.2 Temperate coral (*O. arbuscula*)

The average $\delta^{11}B_{CaCO_3}$ for the temperate coral *O. arbuscula* evaluated in this study (24.12 ± 0.19‰; $n = 3$; Tables 3 and 4) is consistent with previously published values for aragonitic corals (Table 5; see references therein). Generally, aragonitic corals are enriched in $^{11}$B when compared with a theoretical borate $\delta^{11}$B–pH curve (see Figs. 2 and 4). The main vital effect typically used to explain $^{11}$B enrichment in corals, relative to seawater, is an increase in pH at the coral's site of calcification (e.g., Anagnostou et al., 2012; McCulloch et al., 2012; Rollion-Bard et al., 2011b; Trotter et al., 2011; Wall et al., 2016). This hypothesis is supported by in situ measurements of pH using microelectrodes (e.g., Al-Horani et al., 2003; Ries, 2011) and pH-sensitive fluorescent dyes (Venn et al., 2009, 2011, 2013). The $\delta^{11}$B of the coral's skeleton is ~~not sufficiently high so as to be~~ consistent with incorporation of ~~significant~~ boric acid into the coral's aragonite lattice

### 4.2.3 Tropical and temperate urchins (*E. tribuloides*, *A. punctulata*)

The average $\delta^{11}B_{CaCO_3}$ values for the tropical urchin *E. tribuloides* (18.71 ± 0.26‰; $n = 3$; Tables 3 and 4) and the temperate urchin *A. punctulata* (16.28 ± 0.86‰; $n = 3$; Tables 3 and 4) evaluated in this study, which were both reared at equivalent seawater conditions ($pH_{SW} = 8.0$; 25 °C; 32 psu; Table 4), are lower than $\delta^{11}B_{CaCO_3}$ previously reported for other echinoid species (see Table 4; 22.7–22.8‰) but are close to theoretical values of dissolved borate at the same $pH_{SW}$ (17.33‰; Fig. 4). Microelectrode evidence suggests that urchins calcify from fluids with a $pH_{CF}$ and composition similar to that of seawater (Stumpp et al., 2012), which is supported by our observation that urchin $\delta^{11}B_{CaCO_3}$ is similar to $\delta^{11}$B of dissolved borate. The difference between the $\delta^{11}B_{CaCO_3}$ of these two species of urchin and the theoretical value of $\delta^{11}$B for seawater borate (17.33‰) is +1.38‰ for the tropical urchin and −1.05‰ for the temperate urchin – a difference that exceeds their inter-specimen variability (±0.26‰ for the tropical urchin; ±0.86‰ for the temperate urchin, determined as standard deviation, see Table 5). However, ~~the urchins could achieve~~ this deviation in $\delta^{11}B_{CaCO_3}$ ~~by adjusting~~ pH of their calcifying environment ~~by~~ only ± 0.1 unit (e.g., $pH_{CF}$ of 8.1 and 7.9 yield $\delta^{11}$B of calcification site borate of 18.38 and 16.42‰, respectively; see Table 4). Thus, if deviations in urchin $\delta^{11}B_{CaCO_3}$ from seawater borate $\delta^{11}$B indeed reflect urchins' ability to modify ~~the~~ pH at their site of calcification, these modifications appear to be relatively minor (i.e., ± 0.1 pH units) and not always in a direction that favours calcification – consistent with Stumpp et al.'s (2012) observation that urchin biomineralization can occur in cellular compartments where $pH_{CF}$ is lower than that of seawater. The relatively low $\delta^{11}$B of the urchins' tests is not consistent with the hypothesis that significant boric acid is incorporated into the urchins' high-Mg calcite lattice (Fig. 5).

### 4.2.4 Serpulid worm tube (*H. crucigera*)

The average $\delta^{11}B_{CaCO_3}$ for the calcareous tube of the serpulid worm *H. crucigera* evaluated in this study (19.26 ± 0.16‰; $n = 3$; Tables 3 and 4) is close to the theoretical value of $\delta^{11}$B for seawater borate (Fig. 4). The serpulid worm *H. crucigera* produces its calcareous tube from a combination of aragonite and high-Mg calcite (Ries, 2011b). The worm initially produces a slurry of $CaCO_3$ granules in a pair of anterior glands, which ultimately coalesces within a matrix of inorganic and organic components (Hedley, 1956). The samples of *H. crucigera* evaluated in this study were exposed to environmental conditions ($pH_{SW} = 8.1$; 25 °C; 32 psu; Table 4) yielding a theoretical seawater $\delta^{11}B_{B(OH)_4^-}$ of 18.38‰, which is 0.88‰ less than $\delta^{11}B_{CaCO_3}$ measured for this species. Similar to the tropical urchin discussed above, the serpulid worm could generate this divergence in

$\delta^{11}B_{CaCO_3}$ from seawater $\delta^{11}B_{B(OH)_4^-}$ by elevating $pH_{CF}$ by 0.08 units relative to $pH_{SW}$. The relatively low $\delta^{11}$B of the serpulid worm tube is not consistent with significant boric acid incorporation into the worm's calcite and aragonite lattices (Fig. 5). It should be noted that by producing their tubes from a mixture of aragonite and HMC, serpulid worm biomineralization and the resulting $CaCO_3$ matrix is fundamentally different than that of the other marine calcifiers evaluated in this study, which are predominantly monomineralic. To our knowledge, these are the first reported B isotope measurements for serpulid worm tubes and the $\delta^{11}$B values for this mixed mineralogy precipitating organism are not consistent with significant boric acid incorporation into the carbonate lattice (Fig. 5).

### 4.2.5 American oyster (*C. virginica*)

The $\delta^{11}B_{CaCO_3}$ for the American oyster *C. virginica* evaluated in this study (16.03 ‰ ; $n = 1$; Tables 3 and 4) is less than the theoretical value of seawater $\delta^{11}B_{B(OH)_4^-}$ at equivalent $pH_{SW}$ (Fig. 4). Oysters construct their shells of LMC (aragonite during the larval stage) from a discrete calcifying fluid known as the extrapallial fluid (e.g., Crenshaw, 1972), with hemocytes and organic templates playing a potentially important role in crystal nucleation (e.g., Marie et al., 2012; Mount et al., 2004; Weiner et al., 1984; Wheeler, 1992; Wilbur and Saleuddin, 1983). The specimens of *C. virginica* evaluated in this study were grown in seawater conditions ($pH_{SW} = 8.2$; 25 °C; 32 psu; Table 4) that yield a theoretical seawater $\delta^{11}B_{B(OH)_4^-}$ of 19.57 ‰, which is 3.54 ‰ greater than $\delta^{11}B_{CaCO_3}$ measured for this species. The observation that oyster $\delta^{11}B_{CaCO_3}$ is substantially less than seawater $\delta^{11}B_{B(OH)_4^-}$ suggests that the $pH_{CF}$ of oyster extrapallial fluid is less than the pH of the oyster's surrounding seawater. Indeed, pH microelectrode measurements show that the pH of oyster EPF ($pH_{EPF}$) is approximately 0.5 units less than seawater pH, which has been attributed to metabolically driven accumulation of dissolved $CO_2$ when the oyster's shell is closed (Crenshaw, 1972; Littlewood and Young, 1994; Michaelidis et al., 2005). Oysters appear to overcome the low $CaCO_3$ saturation state in the EPF, compared to corals that maintain an elevated $CaCO_3$ saturation state at their site of calcification, by using organic templates to facilitate biomineral growth (e.g., Addadi et al., 2003; Marie et al., 2012; Weiner et al., 1984) and/or maintaining elevated levels of dissolved inorganic carbon within the EPF. The oyster could generate this negative divergence in $\delta^{11}B_{CaCO_3}$ from seawater borate $\delta^{11}$B by decreasing $pH_{CF}$ by 0.35 units (Table 4), which, given the proximity of the independent pH microelectrode measurements of oyster EPF, seems to be a plausible explanation for why oyster $\delta^{11}B_{CaCO_3}$ falls below the theoretical seawater $\delta^{11}B_{B(OH)_4^-}$–pH curve (Klochko et al., 2009; Fig. 5). The relatively low $\delta^{11}$B of the oyster calcite is not consistent with significant boric acid incorpo-

ration into the oyster's calcite lattice (Fig. 5). To the authors' knowledge, these are the first B isotope analyses reported for oysters.

### 4.3 Estimating $pH_{CF}$ from $\delta^{11}B_{CaCO_3}$

The six species of calcifying marine organisms investigated in the present study exhibited average $\delta^{11}B_{CaCO_3}$ ranging from 16.27 to 35.09 ‰ (Table 3). Given that all six species were grown under nearly equivalent controlled laboratory conditions, the large interspecific range in $\delta^{11}B_{CaCO_3}$ supports the hypothesis that $\delta^{11}B_{CaCO_3}$ of biogenic carbonates is not simply inherited from $\delta^{11}B_{B(OH)_4^-}$ of the organism's surrounding seawater (see Table 5 and references therein). Rather, we assert that this species-dependent variability in $\delta^{11}B_{CaCO_3}$ is driven by interspecific differences in the organisms' $pH_{CF}$. To explore this assertion, $\delta^{11}B_{CaCO_3}$ values were converted to $pH_{CF}$ from measured seawater temperature, salinity, a total dissolved boron $\delta^{11}$B value of 39.61 ± 0.20 ‰ (Foster et al., 2010), and an $\alpha$ of 1.0272 (Klochko et al., 2006; Table 4). In the absence of direct measurements of calcifying fluid temperature, salinity, and total dissolved boron $\delta^{11}$B, these parameters are assumed to be equivalent to those of the organism's surrounding seawater. Assuming that only borate is incorporated into the organisms' shells and skeletons (see Table 4), these calculations yield a $pH_{CF}$ (in order of decreasing magnitude) of 9.4 for the coralline red alga (*Neogoniolithion* sp.), 8.5 for the temperate coral (*O. arbuscula*), 8.2 for the serpulid worm (*H. crucigera*), 8.1 for the tropical urchin *(E. tribuloides)*, and 7.9 for the temperate urchin (*A. punctulata*) and American oyster (*C. virginica*).

### 4.3.1 Nonlinearity of the $\delta^{11}B_{CaCO_3}$–$pH_{CF}$ relationship relative to $pK_B$

The determination of $pH_{CF}$ from $pK_B$, $\delta^{11}$B of calcifying fluid ($\delta^{11}B_{CF}$), and $\delta^{11}B_{CaCO_3}$ can be summarized with the following equation (Eq. 1):

$$pH_{CF} = pK_B - \log\left(\left(\delta^{11}B_{CF} - \delta^{11}B_{CaCO_3}\right)/\right.$$
$$\left.\left(\delta^{11}B_{CF} - \left(\alpha \times \delta^{11}B_{CaCO_3}\right) - 1000(\alpha - 1)\right)\right), \quad (2)$$

where $pK_B$ is 8.6152 (at 25 °C and 32 psu; Dickson, 1990), $\delta^{11}B_{CF}$ is 39.61 ‰ (inherited from $\delta^{11}B_{SW}$; Foster et al., 2010), and $\alpha$ is 1.0272 (Klochko et al., 2006). ~~Thus, Eq. (1) allows for the calculation of $\delta^{11}B_{CaCO_3}$ across a range of $pH_{CF}$ (Fig. 1b; Table S1 in the Supplement).~~

The sensitivity of $\delta^{11}B_{CaCO_3}$ to changes in $pH_{CF}$ increases as $pH_{CF}$ approaches $pK_B$ (8.6152; Table S1). For example, a change in $pH_{CF}$ from 7.75 to 7.80 predicts a $\delta^{11}B_{CaCO_3}$ difference of 0.35 ‰ (15.77–15.42 ‰), whereas a change in pH from 8.35 to 8.40 predicts a $\delta^{11}B_{CaCO_3}$ difference of 0.74 ‰ (22.59–21.85 ‰). Thus, the relationship between $pH_{CF}$ and $\delta^{11}B_{CaCO_3}$ is nonlinear over the $pH_{CF}$ range of in-

terest ($7 < \text{pH} < 10$), with pH having the greatest influence on $\delta^{11}$B$_{\text{CaCO}_3}$ as ~~fluid~~ pH$_{\text{CF}}$ approaches pK$_B$.

Fortuitously, the calcifiers investigated in the present study maintain their pH$_{\text{CF}}$ within approximately 1 pH unit of pK$_B$

(i.e., over the interval where small differences in pH$_{\text{CF}}$ cause relatively large differences in $\delta^{11}$B$_{\text{CaCO}_3}$). Therefore, for these organisms, it will be easier to obtain precise measurements of expected differences in $\delta^{11}$B$_{\text{CaCO}_3}$ and, thus, differences in pH$_{\text{CF}}$. Conversely, it will be harder to obtain precise

measurements of the differences in $\delta^{11}$B$_{\text{CaCO}_3}$ (and pH$_{\text{CF}}$) for calcifiers that maintain their pH$_{\text{CF}}$ more distal from pK$_B$ – if such calcifiers indeed exist.

Along these same lines, slight differences in pH$_{\text{SW}}$ of the experimental treatments (also proximal to pK$_B$) could con-

15 ceivably translate to relatively large changes in $\delta^{11}$B$_{\text{B(OH)}_4^-}$ amongst the species' seawater treatments and, thus, their calcifying fluid $\delta^{11}$B$_{\text{B(OH)}_4^-}$ and $\delta^{11}$B$_{\text{CaCO}_3}$. However, the small range of pH$_{\text{SW}}$ for the different species' experimental treatments (8.0–8.2; Table 4) could only account for a

20 2.24‰ range in $\delta^{11}$B$_{\text{CaCO}_3}$ (Table S1), which is far less than the ca. 20‰ range that was observed amongst the different species in the present study. It therefore follows that the large variability in $\delta^{11}$B$_{\text{CaCO}_3}$ (ca. 20‰) observed for the investigated species requires an alternative explanation, such as

~~fundamental~~ differences in their pH$_{\text{CF}}$.

### 4.3.2 Sensitivity of $\delta^{11}$B-derived pH$_{\text{CF}}$ to choice of $\alpha$

As discussed in the introduction (Sect. 1.1), much work has gone into establishing an $\alpha$ that accurately describes the pH-dependent relationship between $\delta^{11}$B of dissolved borate and

30 boric acid in seawater (see Xiao et al., 2014, for detailed discussion), with the earliest published paleo-pH reconstructions using a theoretical value of 1.0194 (Kakihana et al., 1977; see Fig. 2). An empirical $\alpha$ of 1.0272 (Klochko et al., 2006) has now been shown to better predict $\delta^{11}$B$_{\text{B(OH)}_4^-}$,

viz. $\delta^{11}$B$_{\text{CaCO}_3}$, across the range of pH relevant for seawater (Rollion-Bard and Erez, 2010; Xiao et al., 2014). However, $\delta^{11}$B$_{\text{CaCO}_3}$ values of many species of calcifying marine organisms fall either above or below theoretical $\delta^{11}$B$_{\text{B(OH)}_4^-}$–pH$_{\text{SW}}$ curves. It has long been suggested (and shown for

corals) that calcifying organisms diverge from the predicted $\delta^{11}$B$_{\text{CaCO}_3}$ due to their ability to modify the pH of their calcifying environments (e.g., Anagnostou et al., 2012; Hönisch et al., 2004; Krief et al., 2010; McCulloch et al., 2012; Rae et al., 2011; Reynaud et al., 2004; Trotter et al., 2011; Wall et

al., 2016). In the present study, species-specific divergences in $\delta^{11}$B$_{\text{CaCO}_3}$ from the theoretical $\delta^{11}$B$_{\text{B(OH)}_4^-}$–pH$_{\text{SW}}$ curves are interpreted as evidence of the differing capacities of calcifying marine species to modify pH$_{\text{CF}}$. Importantly, existing models of biomineralization for each species are generally

compatible with these $\delta^{11}$B$_{\text{CaCO}_3}$-derived estimates of pH$_{\text{CF}}$ (see Sect. 4.2).

Although an $\alpha$ of 1.0272 (Klochko et al., 2006) was used in the present study to estimate pH$_{\text{CF}}$, other theoretical values for $\alpha$, yielding slightly different $\delta^{11}$B$_{\text{B(OH)}_4^-}$–pH$_{\text{SW}}$ curves (e.g., Byrne et al., 2006; Palmer et al., 1987; see Fig. 4),

will yield slightly different estimates of pH$_{\text{CF}}$ for each organism. For example, using $\alpha$ values of 1.033 (Palmer et al., 1987), 1.0285 (Byrne et al. 2006), 1.0272 (Klochko et al. 2006), and 1.0194 (Kakihana et al. 1977) and a $\delta^{11}$B$_{\text{CaCO}_3}$ of 24.12‰ (temperate coral; pH$_{\text{SW}}$ = 8.1) yields pH$_{\text{CF}}$ values

of 8.7, 8.6, 8.5, and 8.1, respectively – a range of 0.6 pH units. It should also be noted that the lower the $\delta^{11}$B$_{\text{CaCO}_3}$, the more sensitive the reconstructed pH is to choice of $\alpha$. For example, changing $\alpha$ from 1.0272 to 1.0330 will result in a 0.24 pH unit shift for $\delta^{11}$B$_{\text{CaCO}_3}$ of 20‰ but only a 0.12 and

0.08 pH unit shift for $\delta^{11}$B$_{\text{CaCO}_3}$ of 30 and 39.5‰, respectively (see Fig. 4). This underscores the importance of using the same $\alpha$ when comparing $\delta^{11}$B$_{\text{CaCO}_3}$-based estimate of pH$_{\text{CF}}$ amongst species.

### 4.3.3 Implications of $\delta^{11}$B$_{\text{CaCO}_3}$-derived estimates of pH$_{\text{CF}}$ for species-specific vulnerability to ocean acidification

Establishing how marine organisms calcify is a critical requirement for understanding and, ideally, predicting their physiological responses to future ocean acidification (e.g.,

Kleypas et al., 2006). Although it is widely known that many species of marine calcifiers promote calcification by raising pH at their site of calcification, the present study identifies the degree to which this strategy for biocalcification is employed across a range of divergent taxa. Marine calci-

fiers that employ this strategy for calcification may be more resilient to the effects of ocean acidification because their high pH$_{\text{CF}}$ (relative to pH$_{\text{SW}}$) would cause HCO$_3^-$ (elevated due to increased $p$CO$_2$) to dissociate into CO$_3^{2-}$ for calcification, helping the organism to maintain an elevated $\Omega$

at its site of calcification (Ries et al., 2009). Evaluation of this hypothesis in the context of the results of the present study shows that, indeed, the different species' $\delta^{11}$B$_{\text{CaCO}_3}$ and calculated pH$_{\text{CF}}$ exhibit a moderate, inverse relationship with their experimentally determined vulnerability to ocean

acidification (Ries et al., 2009). Species exhibiting more resilient "parabolic" (e.g., coralline red alga) and "threshold" (e.g., coral, tropical urchin) responses to ocean acidification generally exhibited a higher $\delta^{11}$B$_{\text{CaCO}_3}$ and, thus, pH$_{\text{CF}}$ than species exhibiting the more vulnerable "negative" re-

sponses (e.g., oyster, serpulid worm) to ocean acidification (Table 4). The temperate urchin was the exception to this general trend, as it exhibited a relatively resilient parabolic response to ocean acidification yet maintained $\delta^{11}$B$_{\text{CaCO}_3}$ and, thus, pH$_{\text{CF}}$ close to that of pH$_{\text{SW}}$. These results sup-

port the assertion that interspecific differences in pH$_{\text{CF}}$ contribute to marine calcifiers' differential responses to ocean acidification – highlighting the need for future queries into the mechanisms driving boron isotope fractionation, biomin-

eralization, and vulnerability to ocean acidification of marine calcifying organisms.

## 5   Conclusions

This study establishes the methodology for measuring stable boron isotopes at Ifremer (Plouzané, France) and reveals that neither cleaning protocol (oxidized vs. untreated), nor method of sample preparation (batch vs. column), nor injection system ($d$-DIHEN vs. ammonia addition) causes a significant difference in the measured $\delta^{11}$B$_{CaCO_3}$ of the evaluated samples and standards. The batch method of boron extraction is identified as preferable to the column chemistry method because the risk of B contamination is reduced in the batch method due to shorter exposure to potential contaminants and smaller reagent volumes.

This newly established method for measuring stable boron isotopes at Ifremer was used to measure the $\delta^{11}$B$_{CaCO_3}$ of six species of marine calcifiers that were all grown under equivalent seawater conditions. The coralline red alga *Neogoniolithion* sp. ($35.89 \pm 3.71$‰; $n = 3$) exhibited the highest $\delta^{11}$B$_{CaCO_3}$, followed by the temperate coral *O. arbuscula* ($24.12 \pm 0.19$‰; $n = 3$), the tube of the serpulid worm *H. crucigera* ($19.26 \pm 0.16$‰; $n = 3$), the tropical urchin *E. tribuloides* ($18.71 \pm 0.26$‰; $n = 3$), the temperate urchin *A. punctulata* ($16.28 \pm 0.86$‰; $n = 3$), and the temperate oyster *C. virginica* ($16.03$‰ ; $n = 1$). The observed ca. 20‰ range in $\delta^{11}$B$_{CaCO_3}$ composition of the investigated species constitutes the largest range in biogenic $\delta^{11}$B$_{CaCO_3}$ reported to date.

Consideration of these extreme interspecific differences in $\delta^{11}$B$_{CaCO_3}$ in the context of existing models of biomineralization for the investigated species, combined with published measurements of pH$_{CF}$ for some of the species, generally supports the assertion that most marine calcifiers precipitate their CaCO$_3$ from a discrete calcifying medium with a pH that is either greater than, equivalent to, or, for some species, less than external seawater pH. Furthermore, the observation that the different species' $\delta^{11}$B$_{CaCO_3}$ and calculated pH$_{CF}$ generally varied inversely with their experimentally determined vulnerability to ocean acidification suggests that a species' relative resilience (or vulnerability) to OA may be influenced by their ability (or lack thereof) to maintain an elevated pH$_{CF}$. These observations contribute to the growing body of work that uses $\delta^{11}$B$_{CaCO_3}$ as a tool to advance understanding of the mechanisms by which marine calcifiers build and maintain their shells and skeletons and, ultimately, how these organisms will respond to anthropogenic CO$_2$-induced ocean acidification.

*Data availability.*  . TS7

The Supplement related to this article is available online at https://doi.org/10.5194/bg-15-1-2018-supplement.

*Author contributions.*  RAE and JBR conceived of the project and wrote the proposals that funded the work. JBR performed the culturing experiments. RAE, JNS, and JBR contributed to the experimental design. JNS, YWL, MG, EP, and RAE contributed to developing the method of boron isotope analysis. JNS performed the measurements with assistance from EP. JNS conducted the data analysis. Interpretation was led by JNS and RAE with input from JBR and YWL. JNS drafted the paper, which was edited by all authors. This is publication no. 361 from Northeastern's Marine Science Center.

*Competing interests.*  The authors declare that they have no conflict of interest. TS8

*Acknowledgements.*  This work was supported by the Laboratoire d'Excellence LabexMER (ANR-10-LABX-19) and co-funded by a grant from the French government under the programme Investissements d'Avenir, as well as by a grant from the Regional Council of Brittany (SAD programme). Robert A. Eagle and Justin B. Ries also acknowledge support from National Science Foundation grants OCE-1437166 and OCE-1437371. We thank J.-P. D'Olivo TS9 and the members of the UWA lab for supplying us with an aliquot of the NEP standard.

Edited by: Xinming Wang
Reviewed by: Jesse Farmer and two anonymous referees

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

## Remarks from the language copy-editor

CE1  Should this be "Math Sciences..."?

CE2  Please verify that this definition is correct.

CE3  Should this be "data sets"?

CE4  Please verify that this definition is correct.

CE5  Please verify that this is correct.

CE6  Please verify that this definition is correct.

CE7  This sentence is incomplete. Please rephrase.

CE8  What corresponds to what? This is unclear. Please clarify.

CE9  You haven't used NA in this table.

## Remarks from the typesetter

TS1  Please check email address. It differs from our system (jill-naomi.sutton@univ-brest.fr).

TS2  Copernicus Publications collects the DOIs of data sets, videos, samples, model code, and other supplementary/underlying material or resources as well as additional outputs. These assets should be added to the reference list (author(s), title, DOI, and year) and properly cited in the article. If no DOI can be registered, assets can be linked through persistent URLs. This is not seen as best practice and the persistence of the URL must be secured.

TS3  The composition of all figures has been adjusted to our standards.

TS4  Please confirm change.

TS5  Please confirm superscription in the equation above.

TS6  Please confirm here and throughout the text?

TS7  Please provide a statement on how your underlying research data can be accessed. If the data are not publicly accessible, a detailed explanation of why this is the case is required. The best way to provide access to data is by depositing them (as well as related metadata) in reliable public data repositories, assigning digital object identifiers (DOIs), and properly citing data sets as individual contributions. Please indicate if different data sets are deposited in different repositories or if data from a third party were used. If no DOI is available, assets can be linked through persistent URLs to the data set itself (not to the repositories' home page). This is not seen as best practice and the persistence of the URL must be secured.

TS8  Declaration of all potential conflicts of interest is required by us as this is an integral aspect of a transparent record of scientific work. If there are possible conflicts of interest, please state what competing interests are relevant to your work.

TS9  Please provide full author name.

TS10  Please provide page range.

TS11  Please provide page range and DOI number.

TS12  Please check page range.

TS13  Please cehck volume.

TS14  Please provide full journal name.

TS15  Please provide page range.

TS16  Please provide page range.

TS17  Please provide page range.

TS18  Please check link and provide last access date. Link can't be found.

TS19  Please provide page range.

TS20  Please provide journal name.

TS21  Please provide page range.

TS22  Please provide page range.