# Peer review of "$\delta^{11}$ B as monitor of calcification site pH in divergent marine calcifying organisms"

_Biogeosciences, 2017_

## Referee Comment (RC1) · Anonymous Referee #1 · 8 Jun 2017

This paper presents the boron isotopic composition of 6 different marine calcifying organisms (both aragonite and calcite), and suggests that divergent 11B values of each species result from different pH values in their calcifying fluid, as these organisms can physiologically regulate the composition of their calcifying fluid. Generally, the results are robust and interesting, but do not offer any new or significant explanation and insights into the current knowledge of 11B of marine carbonates.

My biggest concern for the manuscript is the pH calculation for calcite 11B, for instance the coralline algae, urchin and oyster. The assumption of the 11B-pH calculation is based on that only borate ion enters into the mineral lattice. But as confirmed by recent theoretical calculation (ab initio), and NMR and X-ray spectromicroscopy experiments (Balan et al., 2016, GCA; Branson et al., 2015, EPSL; Mavromatis et al.,

2016, GCA;Noireaux et al., 2015, EPSL), trigonal B also incorporates into the lattice (especially for calcite) to substitute $CO_3^{2-}$ in a large proportion. Therefore, the derived pH from calcite 11B is questionable and may bias the interpretation of the data. I think the authors should mention this in their ms to open the mind of readers that there are other possibilities to interpret marine carbonate 11B.

Moreover, numerous literature works have confirmed that 11B-derived pH reflect the pH of calcifying fluid (or extracellular body fluid) instead of ambient seawater (Allison et al., 2014; Georgiou et al. 2015, PNAS; Trotter et al., 2011, EPSL; McCulloch et al., 2012, 2017; GCA; Heinemann et al., 2012, G3), though most of these results are for aragonitic calcifiers. It's therefore not surprising that boron isotope compositions in different calcifying organisms or even different species are different, since the ability of calcifying organisms to regulate their calcifying fluid pH may vary among taxa. Furthermore, although the calcification mechanism for each calcifier remains largely unknown, a brief introduction of current understanding would be helpful for the readers to understand the relationship between 11B and calcifying fluid pH, and between the internal pH and seawater pH.

Specifically,

Lines 42-43, some of the references cited here are not related to boron isotopes, e.g. Saenger et al., 2013; Zinke et al., 2014.

Lines 46-47 "11B composition of borate in seawater (11BSW; Pagani et al.,2005) " 11Bsw is commonly used to indicate boron isotope composition of seawater other than that of borate. Please modify.

Line 59, "The 11B of modern seawater is 39.61 ± 0.04 ‰ (Foster et al., 2010)", 2SD should be used when reporting a replicated and certified value, so the value should be 39.61± 0.20 ‰ (Foster et al., 2010).

Lines 164-168: The setup of culturing experiment should be illustrated in details in the

Methods and Materials Section, as well as the method to identify and separate the new growth part of each skeleton or shell for isotope measurement.

Lines 168-171 This is a replicate of Section 2.3, and not an important part for the Introduction, so please remove.

Line 279 Is the analytical precision shown here 2SD or SD? Lines 285-286 Please remove this section.

Lines 333-334 "polymorph mineralogy was not found to influence boron isotope fractionation (Noireaux et al. 2015)." This seems to be in contrary to the conclusion of Noireaux et al. 2015 who claim that "Our results indicate that the main controlling factors of 11B are the solution pH and the mineralogy of the precipitated carbonate mineral".

Lines 336-337 "if shell mineralogy was the primary driver of the observed interspecific variation in $\delta$11BCaCO3 compositions – a trend that is not observed". As suggested by Noireaux et al. 2015, both solution pH and mineralogy are important factors controlling 11B in carbonates, differences in calcifying fluid pH may also obscure "this mineralogical trend", especially the underlying calcification mechanisms of each calcifier remain largely unknown. So, I don't think mineralogical influences can be easily excluded.

Lines 368-370 With such high proportion of trigonal B incorporation, the classic 11B-pH equation cannot be used to calculate the pH, as 11Bcarb = 11Bborate is the basic assumption for the calculation.

Line 426 but also mineralogy dependent

Lines 427-432 The premise of using the equation mentioned in the paper to calculate pH by carbonate 11B is that borate ion is the only species that enters into the lattice (for example in aragonite). As suggested by both theoretical calculation and NMR experiment, both boric acid and borate ion exist in the lattice of calcite. Therefore, for those calcite organisms, this 11B-pH equation may not be applicable, and the calculated pH

value may not reliably reflect the pH of calcifying fluid.

---

## Referee Comment (RC2) · J. Farmer (Referee) · 14 Jun 2017

General Comments

Sutton *et al.* present boron isotopic composition measurements of marine calcifying organisms cultured under controlled conditions using modifications to existing MC-ICP-MS measurement procedures. These are interesting results worthy of publication after revision. In addition to the detailed comments below, I have three general recommendations.

1) Discussion of sample collection, subsampling, and what the different isotope measurements were measured on. Namely, there isn't any information, so it is impossible to tell whether multiple specimens were used, whether each specimen was subsampled in the same skeletal region, etc. This must be included in an expanded Materials/Methods section.

2) Wording of boron isotope differences in the study. This study devotes much attention to the fractionation between boric acid and borate in aqueous solution, termed the fractionation factor ($\alpha_B$). However, the study also confusingly defines their differences between the $\delta^{11}$B of carbonates and their expected $\delta^{11}$B based on the $\delta^{11}$B of borate ion in solution as "fractionations".

This is an unnecessary complication. The data of this study do not address the actual fractionation factor ($\alpha_B$), which has been determined, but rather address how carbonate $\delta^{11}$B values may be offset from the $\delta^{11}$B of borate ion in solution. Put another way, this study tests the assumed model that carbonate $\delta^{11}$B records seawater pH via sole incorporation of borate ion from seawater at the measured seawater pH. Carbonate data that are discordant with this model (as in this study) do not necessarily imply any isotopic fractionations; instead, they suggest that one of the assumptions of the model may be wrong when applied to the carbonate in question.

3) The discussion of factors influencing the quantification of calcifying fluid pH needs to be refocused/expanded. The major strength of this study is that all calcifiers were grown under approximately the same conditions. This experimental design effectively minimizes uncertainty arising from variations in $pK_B{}^\star$ and $\alpha_B$. However, the only sources of uncertainty to $pH_{cs}$ discussed are $pK_B{}^\star$ and $\alpha_B$, exactly those that are best controlled. This discussion needs to be expanded to evaluate the effects of known modifications to each carbonate's microenvironment and the possibility of alternate boron incorporation pathways other than borate (particularly for the coralline alga).

Specific comments/Technical corrections:

L15: Suggest change opening sentence to "The boron isotopic composition ($\delta^{11}$B) of marine biogenic carbonates..." and remove $\delta^{11}$B reference on L18

L33/34: Cite original studies of instrumental pH records in lieu/addition to IPCC; e.g., BATS (Bates, 2007); ESOTC (Gonzalez-Davlia et al., 2010), and ALOHA (Dore et al., 2009) or more recent studies

L45-50: The theoretical model for boron incorporation predates Pagani et al. (2005). Please cite original studies; e.g. Hemming and Hanson (1992) for $CaCO_3$ $\delta^{11}B$ reflecting $\delta^{11}B$ borate, and Zeebe and Wolf-Gladrow (2001) for description of parameters needed to calculate pH from $\delta^{11}B$.

L46-47: The definition of $\delta^{11}B_{sw}$ is misleading; $\delta^{11}B_{sw}$ is the isotopic composition of boron in seawater (e.g., L49), which reflects the sum of all boron species in seawater. However, on L46 the text states the boron isotopic composition of borate in seawater. This is not $\delta^{11}B_{sw}$, but instead is defined separately ($\delta^{11}B_{borate}$ or similar) and is a function of both $\delta^{11}B_{sw}$ and pH.

L58: Catanzaro et al. (1970): Boric Acid: Isotopic and Assay Standard Reference Materials is the appropriate reference for NIST 951.

L59: "constant" instead of "consistent"

L64: "was" instead of "has been". Since the identification of errors in Kakihana's vibrational spectra (L69), the Kakihana fractionation is not appropriate and is not used for $\delta^{11}B$-pH applications. Hönisch et al. (2007) (response to Pagani et al., 2005) discusses the fractionation and the concept of species-specific calibrations in more detail.

L72-76: Reword this. The studies referenced on L75-76 do not argue for different fractionation factors; rather, they argue for species-specific calibrations between $\delta^{11}B_{CaCO3}$ and seawater pH. I am unaware of any evidence that the isotopic fractionation between boric acid and borate (the fractionation factor) is fundamentally different in biogenic calcifying fluids than in seawater. Moreover, any insights of calcifying fluid pH require assuming that the same fractionation factor applies in both seawater and calcifying fluid (e.g., Trotter et al., 2011).

Section 1.1 could use greater clarity for pH terms. The manuscript starts with discussing seawater pH, and $\delta^{11}$B as a seawater pH proxy, but transitions to calcifying fluid pH in this section. For the sake of readability, I suggest you define these separately here (seawater pH = pH$_{sw}$ and calcifying fluid pH = pH$_{cf}$, or similar), and use throughout the text.

L83-85: This is true, although you could additionally cite several recent reinterpretations of boron incorporation into carbonates (Norieaux et al., 2015; Uchikawa et al., 2015, Balan et al., 2016).

L94-95: Suggest rewording to "organisms' ability to regulate pH at their site of calcification" L108: Ca$^{2+}$

L109-125: This would be better placed at the beginning of Section 1.2, before the discussion of OA reducing [CO$_3^{2-}$]

L125: Specify that K$^\star_{sp}$ is a function of temperature

L136: Is there a section 1.3? If not, change this to "1.3"

L137-138: Rephrase to "may record" and cite studies suggesting that carbonate $\delta^{11}$B records calcifying fluid pH. While it has been hypothesized that $\delta^{11}$B records pH in the calcifying microenvironment, to say that $\delta^{11}$B$_{CaCO3}$ should record calcifying microenvironment pH is a stretch given current uncertainties in how pH is controlled in these microenvironments (e.g., Section 1.2), and uncertainties in the $\delta^{11}$B proxy (see discussion in Farmer et al., 2015).

L164-171: Move to Methods section as a "Materials" subsection. Also, how were the specimens subsampled for isotopic analysis? Were they bulk homogenized or subsampled on particular growth features? Is there only one specimen per taxa or multiple specimens? This is very important to include as it might shed light on some of the poor reproducibility you observe (especially for the coralline alga).

L198-200: You can remove the sentence starting with "All samples. . ." since you discuss this immediately below.

L209: Were the samples just rinsed in buffered UHQ water, or were they stored in the water? It is unclear if they were then acidified in this water medium (or not).

L224 (Batch method): Can you specify the type of microcentrifuge tubes used (polypropylene vs. PFA/Teflon), and whether the tubes were reused between samples? Or did you transfer the resin into separate microcentrifuge tubes for each sample? If the latter, how did you store the resin between samples?

L252: "The $\delta^{11}$B was also evaluated..." Please explain. This reads as if you used the internal carbonate standards to correct your $\delta^{11}$B values, which would not be appropriate. As these are internal standards, do you mean to say that you are using them to evaluate the efficacy of the preparation and measurement protocol?

L285-286: Unnecessary subsection; please remove.

L308-322: Based on Table 3, it seems that batch separation with NH3 injection was most commonly used. Does that reflect your experience with the different methods and which one seemed most replicable and user friendly? Can you make a recommendation on which separation and injection methodology you think other others should follow?

L320: Can you comment on how much lower the batch method blanks were? The procedural blanks are listed as sub-nanogram (L235), but I cannot find a distinction between column vs. batch protocols.

L334-360: Multiple references to a phantom Figure 5. Please check Figure references throughout text.

L360 (Coralline red alga): Please comment on why the $\delta^{11}$B values for these specimens are so different ($\pm 3.7$ per mil uncertainty is massive!).

L368-372. Couldn't both be possible-e.g., microenvironment pH adjustment and boric

acid incorporation? If that was the case, could you actually determine the pH at the site of calcification? I'd strongly recommend including a figure and calculations showing how the derived value of pHcs would change as a function of varying % boric acid incorporation.

Moreover, $pH_{cs}$=9.4 seems pretty extreme. Is there any evidence for a physiological advantage to a calcifying organism obtaining such alkaline pH in its calcifying medium? I don't disagree with the proposed mechanism (algal photosynthesis), just the magnitude. I imagine that at this pH, CaCO3 would spontaneously precipitate (due to massively high omega), which would not be desirable for the organism.

Finally, note that NMR is not useful for quantifying % boric acid incorporation (see and reference Balan et al., "First-principles study of boron speciation in calcite and aragonite" GCA 193, 2016).

Section 4.3: Seems out of order. I find it more intuitive to present the equation for pH calculation first (L455-460), then discuss/test the assumptions of this approach (Section 4.3.2), then finally loop around to the best estimate of $pH_{cs}$ and comparison to OA responses.

L426: This is not the correct terminology. This study's data do not suggest that the B isotope fractionation is species dependent; there is no direct measurement of the fractionation in this study. Rather, these data suggest that the B isotope composition of these taxa cannot be explained solely by borate incorporation at ambient seawater pH.

L430: Specify the assumption that only borate is incorporated here

L449: "by testing the factors that may influence the theoretical model of borate $\delta^{11}$B variation as a function of pH"

The theoretical model of carbonate $\delta^{11}$B reflecting seawater pH has three parts: 1) borate $\delta^{11}$B varies with pH in a known fashion (requires knowledge of pH, $pK_B$* as you say

here); 2) carbonates are calcified from unmodified seawater; 3) boron in carbonates results from solely borate incorporation. Only by combining all three assumptions can you use carbonate $\delta^{11}$B to record seawater pH. You've discussed 1; please discuss what your data suggest about 2 and 3, and how uncertainties in these assumptions could influence your data. You've already discussed 2 (modification of calcification site chemistry) throughout the discussion; bring it all together here.

L475-495: The first paragraph (L475-486) is just rehashing the introduction and can be removed. The second paragraph (L487-495) is the meat of this.

L478: Figure 2 instead of Figure 3?

L499: I would urge caution with this relationship between pH elevation and OA response, as it is at best a qualitative relationship. Also, seeing as this is a central point of the manuscript, I would recommend including a figure to illustrate the relationship between pH elevation and OA response (something like Doney et al. 2009's Figure 4 may work). L505: Please note that this "species-specific" calibration approach is not new; it has been the standard procedure in the boron isotope community for years, as demonstrated by numerous studies that should be cited here (e.g., Sanyal et al., 1996; 2001; Hönisch et al., 2003; Trotter et al., 2011; Anagnostou et al., 2012; etc.)

Figure 2: -The seawater borate curves must be mislabeled; all else being equal, increasing alpha will lead to a lower $\delta^{11}$B-borate at lower pH. I think the dotted line should be Kakihana and the solid line should be Klochko (see also Fig. 4).

-Why did you choose to plot these specific data? The chosen ones seem quite random, and there are many other data out there worth considering (as your Table 5 illustrates) that may be most appropriate for comparison with the carbonates presented in this study.

-Is the large pH range on the x-axis (7-10) necessary? It is difficult to make out the individual studies.

-Note also typos: "Hönisch", "Brachiopod" and "Penman".

Figures 2 and 4: Please also plot $\delta^{11}$B – boric acid for the fractionations.

---

## Referee Comment (RC3) · Anonymous Referee #3 · 26 Jun 2017

The manuscript by Sutton et al reports the boron isotope compositions of various marine calcifyers (coralline red alga, urchins, worm, coral, oyster). All the samples came from culture experiment (T=25°C, pCO2=409 $\mu$atm) and so should record the same $\delta$11B values if no vital effects are present. The $\delta$11B range of all the data is about 20‰ and seems to show the biological control on the calcification pH. I found the data interesting, but I think that there are a lot of repetitions through the text. Even if it is mentioned in the case of coralline red alga, the influence of B3 is not really taken into account. For example, the presence of B3 was also shown in corals, and it was not described in the text. In the figures, the symbols should be different between the calcium carbonate polymorphs.

More technical comments: -L53: B(OH)4- -L85: please mention the study of Noireaux

et al (2015) -L108, 127: "2" must be in superscript -L142: please mention the study of Noireaux et al (2015) -L151-152: Please mention the studies of Rollion-Bard et al (2003, 2011) -L154: Please mention the study of Jorgensen et al (1985) -L221: Interest for what? Why the data are not shown in the manuscript? -L243-244: It was already mentioned, please delete -L256: I suppose that there are older references than McCulloch et al (2014) for the MC-ICP-MS method. - section 3.1.1.: Do you have an idea why the measurements on JCt-1 are more variable? -L288, 324: please add the errors on the $\delta$11B values -L290: Why the error on the $\delta$11B value of the coralline alga is so high? -L327: "range in range", please correct -L334: No, in Noireaux et al (2015) there is a clear effect of the mineralogy (see figure 1) -L358: please remove the part of the sentence concerning boron isotopes. In this sentence, it is explained that there is an enrichment of 11B in corals and that it is supported by 'boron isotope analyses' (of course!). -L370-371: What would be the pH of calcification if there is effectively 30% of B3? The $\delta$11B value of coralline alga could result from the combination of a pH increase and the incorporation of a certain proportion of B3. -section 4.2.3: What are the calculated pH if the results of Noireaux et al (2015) for inorganic calcite are taken into account? -L402: please remove 'Notably....worm tubes' -L420: Klochko et al (2006), instead of Klochko (2009) -L420: please remove 'Notably....oysters' -L477: Kakihana et al (1977) instead of Kakihana (1977) -L495: It is obvious. I do not see the point here. -section 4.3.3: It was already mentioned, please delete this section

Table 3: 'JCp-1' instead of 'JCP-1'.

Figure 1: Please use the alpha of Klochko et al (2006) and specify in the caption the pKa used and the alpha used. Figure 2: Please add data of Reynaud et al (2004), Lécuyer et al (2002), Farmer et al (2005). Please use the full name species of the foraminifera. 'Brachiopod' instead of 'Brochiopod'; 'Penman' instead of 'Penmen'.

Table S1: In the caption, specify the pKa and alpha used.

---

## Author Response (AR1)

**Reply to reviewer 1**

We wish to thank the reviewer for their succinct and thoughtful analysis of our manuscript and criticisms concerning the calculation of pH for calcite $\delta^{11}B$.

Our intention with this manuscript was to consider calcification site pH as the primary driver of the observed range in $\delta^{11}B$ values for the evaluated species as a primary hypothesis, and to discuss whether the values we obtained made sense given what else is known about their bio-calcification strategies including independent measurements of internal pH from other methods (eg. pH sensitive dyes, microelectrodes). Our intention was to set up this hypothesis as a straw man to see whether our data were constant with the idea of primary pH control across these diverse species, and if it was not then to discuss other factors that could be at play. At a at number of points in the manuscript we also explore possible alternative explanations including mineralogical effects, boric acid incorporation, and we also discuss uncertainties in the calculation of an absolute pH value from carbonate $\delta^{11}B$ values.

We agree with the reviewer that it is possible that there are additional complications to the interpretations in calcitic organisms. In particular, the possible incorporation of trigonal boron/boric acid into the skeleton and many of the studies mentioned are already referenced in the text. This was addressed to a degree in the original text. For example, lines 329-331 we state "Boron co-precipitation with inorganic CaCO3 (i.e. abiogenic) is known to be dependent on solution pH and inorganic CaCO3 precipitation rate, however, the relative abundances of the inorganic B species in solution that are incorporated into inorganic CaCO3 (borate ion and boric acid) have been shown to be independent of the parent solution pH (Mavromatis et al. 2015)." and also lines 365-370 where we state "…raising the possibility that coralline red alga incorporate both species of dissolved inorganic boron during calcification. In support of this argument, Cusack et al. (2015) provide NMR data indicating that 30 % of the B incorporated into the coralline red alga *Lithothamnion glaciale* was present as boric acid. However, since the coralline red algae were reared at a pH of 8.1, the $\delta^{11}B_{CaCO3}$ compositions observed for the coralline alga in the present study would require incorporation of both inorganic species of boron at $[B(OH)_3]:[B(OH)_4-]$ ratios of *ca.* 75:25, not the 30:70 ratio observed by Cusack et al. (2015)."

Nevertheless we agree that this information could be more prominently addressed in the main text and propose that a series of changes could be included to address this alternative hypothesis that particularly affects the calcite mineralizing organisms. For example, we can add a clarifying sentence in the abstract and provide more discussion to section 4.2.

We believe an interesting aspect of our study is that of the three calcitic organisms analyzed, only one of them has $\delta^{11}B$ values (the coralline alga, $35.89 \pm 3.71$ ‰; n = 3) potentially consistent with boric acid incorporation, as suggested in *Noireaux et al., 2015, EPSL, Balan et al., 2016, GCA; Branson et al., 2015, EPSL; Mavromatis et al., 2016, GCA.* This possibility has also been explored in a NMR based study of coralline algae by Cusack et al. 2015 and we did cite this study on lines 366-370 and provide some discussion, as mentioned above. Nevertheless we agree with the reviewer that aspects of the text could be tightened up in this respect (for example clarifying the findings of Noireaux et al., 2015 as the reviewer highlighted).

The high-Mg calcite urchin species we studied have low $\delta^{11}B$ values, with one species even having a value that is actually lower than the seawater borate $\delta^{11}B$ value (the temperate

urchin *A. punctulata*; $16.28 \pm 0.86$ ‰; n = 3; Tables 3 and 4),). Given these values in the two species of urchin-examined incorporation of significant amounts of seawater boric acid with much higher $\delta^{11}B$ values, seems unlikely.

In our manuscript we do have a separate section discussing biomineralization strategies of each organism studied (sections 4.2.1 to section 4.2.5), however in response to the reviewer's comment we can certainly expand this.

We also accept that there are of course many previous studies reporting a range of $\delta^{11}B$ values in marine organisms. However we believe a striking finding of our study is the extreme range (20 per mil) we observed in species that were cultured in environmentally controlled and equivalent conditions.

*Lines 42-43, some of the references cited here are not related to boron isotopes, e.g. Saenger et al., 2013; Zinke et al., 2014.*

Author Response:
Originally these references were meant to indicate that corals have disequilibrium "vital" effects in other isotope systems but they can be removed from this sentence since it does not make sense to include them now.

*Lines 46-47 "11B composition of borate in seawater (11BSW; Pagani et al.,2005) " 11Bsw is commonly used to indicate boron isotope composition of seawater other than that of borate. Please modify.*

Author Response:
This should read *"$\delta^{11}B$ composition of borate in seawater ($\delta^{11}B_{(OH)4}$; Pagani et al.,2005)"* and will be changed.

*Line 59, "The 11B of modern seawater is $39.61 \pm 0.04$ ‰ (Foster et al., 2010)", 2SD should be used when reporting a replicated and certified value, so the value should be $39.61 \pm 0.20$ ‰ (Foster et al., 2010).*

Author Response:
We will fix this.

*Lines 164-168: The setup of culturing experiment should be illustrated in details in the Methods and Materials Section, as well as the method to identify and separate the new growth part of each skeleton or shell for isotope measurement.*

Author Response:
We can certainly add more information on the culturing setup and information on the method used for new growth identification. To clarify, the cultures presented here were previously reported in another publication, Ries et al., 2009, so we feel it is appropriate to summarize important elements of the culturing setup rather than repeat the very detailed description in that paper.

*Lines 168-171 This is a replicate of Section 2.3, and not an important part for the Introduction, so please remove.*

Author Response:
We can remove this.

*Line 279 Is the analytical precision shown here 2SD or SD?*

Author Response:
The analytical precision shown here is 2SD, this will be mentioned in the text.

*Lines 285-286 Please remove this section.*

Author Response:
We can remove this.

*Lines 333-334 "polymorph mineralogy was not found to influence boron isotope fractionation (Noireaux et al. 2015)." This seems to be in contrary to the conclusion of Noireaux et al. 2015 who claim that "Our results indicate that the main controlling factors of 11B are the solution pH and the mineralogy of the precipitated carbonate mineral".*

Author Response:
This is a glaring error on our part. We tried to simplify an argument, and the message was lost in translation. Thank you for picking up on this.

The sentence should read "Although Mavromatis et al. (2015) also found that polymorph mineralogy influences both the B/Ca ratio (higher in aragonite than calcite) and speciation of B in inorganic CaCO3 (borate/boric acid ratio higher in aragonite than calcite), B incorporation alone does not appear to influence boron isotope fractionation.

*Lines 336-337 "if shell mineralogy was the primary driver of the observed interspecific variation in δ11BCaCO3 compositions – a trend that is not observed". As suggested by Noireaux et al. 2015, both solution pH and mineralogy are important factors controlling 11B in carbonates, differences in calcifying fluid pH may also obscure "this mineralogical trend", especially the underlying calcification mechanisms of each calcifier remain largely unknown. So, I don't think mineralogical influences can be easily excluded.*

Author Response:
We have somewhat touched on this in the points above. We agree that the possibility of boric acid incorporation needs to be discussed. The coralline alga has a $\delta^{11}B$ value potentially consistent with the mineralogical effect discussed, but not the urchin species.

*Lines 368-370 With such high proportion of trigonal B incorporation, the classic 11BpH equation cannot be used to calculate the pH, as 11Bcarb = 11Bborate is the basic assumption for the calculation.*

Author Response:
As we believe data from only one of our species is potentially (but not necessarily) consistent with significant trigonal B incorporation relative to borate (see responses above) we prefer to discuss this as uncertainty when calculating calcification site pH, adopting the straw man approach we described above.

*Line 426 but also mineralogy dependent*

Author Response:
We can add a qualifying addition here.

*Lines 427-432 The premise of using the equation mentioned in the paper to calculate pH by carbonate 11B is that borate ion is the only species that enters into the lattice (for example in aragonite). As suggested by both theoretical calculation and NMR experiment, both boric acid and borate ion exist in the lattice of calcite. Therefore, for those calcite organisms, this 11B-pH equation may not be applicable, and the calculated pH value may not reliably reflect the pH of calcifying fluid.*

Author response:
This point is largely covered in our response to previous comments.

**Reply to reviewer 2**

We wish to thank Dr. Jesse Farmer for their thorough review of our manuscript and their helpful comments. We believe that we can address all of the major comments indicated by Dr. Farmer as indicated in the discussion below.

*1) Discussion of sample collection, subsampling, and what the different isotope measurements were measured on. Namely, there isn't any information, so it is impossible to tell whether multiple specimens were used, whether each specimen was subsampled in the same skeletal region, etc. This must be included in an expanded Materials/Methods section.*

Author response:
We will include this information.

*2) Wording of boron isotope differences in the study. This study devotes much attention to the fractionation between boric acid and borate in aqueous solution, termed the fractionation factor (_B). However, the study also confusingly defines their differences between the _11B of carbonates and their expected _11B based on the _11B of borate ion in solution as "fractionations".*

*This is an unnecessary complication. The data of this study do not address the actual fractionation factor (_B), which has been determined, but rather address how carbonate _11B values may be offset from the _11B of borate ion in solution. Put another way, this study tests the assumed model that carbonate _11B records seawater pH via sole incorporation of borate ion from seawater at the measured seawater pH. Carbonate data that are discordant with this model (as in this study) do not necessarily imply any isotopic fractionations; instead, they suggest that one of the assumptions of the model may be wrong when applied to the carbonate in question.*

Author response:
We can address this comment as discussed below in the responses to more specific comments.

*3) The discussion of factors influencing the quantification of calcifying fluid pH needs to be refocused/expanded. The major strength of this study is that all calcifiers were grown under approximately the same conditions. This experimental design effectively minimizes uncertainty arising from variations in pKB\* and _B. However, the only sources of uncertainty to pHcs discussed are pKB\* and _B, exactly those that are best controlled. This discussion needs to be expanded to evaluate the effects of known modifications to each carbonate's microenvironment and the possibility of alternate boron incorporation pathways other than borate (particularly for the coralline alga).*

Author response:
We can address this complication as discussed below in the responses to more specific comments.

More specific comments

*L15: Suggest change opening sentence to "The boron isotopic composition (_11B) of marine biogenic carbonates: : :" and remove _11B reference on L18*
*L33/34: Cite original studies of instrumental pH records in lieu/addition to IPCC; e.g., BATS (Bates, 2007); ESOTC (Gonzalez-Davlia et al., 2010), and ALOHA (Dore et al.,2009) or more recent studies*

*L45-50: The theoretical model for boron incorporation predates Pagani et al. (2005). Please cite original studies; e.g. Hemming and Hanson (1992) for CaCO3 _11B reflecting _11B borate, and Zeebe and Wolf-Gladrow (2001) for description of parameters needed to calculate pH from _11B.*

Author response to above comments:
We will change this information as recommended by the reviewer.
We will cite Byrne et al. (2010); Vázquez-Rodrıguez et al. (2012) in L33, Feely et al. (2016); Feely et al. (2008) in L34

*L46-47: The definition of _11Bsw is misleading; _11Bsw is the isotopic composition of boron in seawater (e.g., L49), which reflects the sum of all boron species in seawater. However, on L46 the text states the boron isotopic composition of borate in seawater. This is not _11Bsw, but instead is defined separately (_11Bborate or similar) and is a function of both _11Bsw and pH.*

Author response to above comments:
This should read "$\delta^{11}B$ composition of borate in seawater ($\delta^{11}B_{(OH)4}$; Pagani et al.,2005)" and will be changed.

*L58: Catanzaro et al. (1970): Boric Acid: Isotopic and Assay Standard Reference Materials is the appropriate reference for NIST 951.*

*L59: "constant" instead of "consistent"*

*L64: "was" instead of "has been". Since the identification of errors in Kakihana's vibrational spectra (L69), the Kakihana fractionation is not appropriate and is not used for _11B-pH applications. Hönisch et al. (2007) (response to Pagani et al., 2005) discusses the fractionation and the concept of species-specific calibrations in more detail.*

Author response to above comments (L58-L64):
We will change *"was" instead of "has been"* as recommended by the reviewer.

*L72-76: Reword this. The studies referenced on L75-76 do not argue for different fractionation factors; rather, they argue for species-specific calibrations between _11BCaCO3 and seawater pH. I am unaware of any evidence that the isotopic fractionation between boric acid and borate (the fractionation factor) is fundamentally different in biogenic calcifying fluids than in seawater. Moreover, any insights of calcifying fluid pH require assuming that the same fractionation factor applies in both seawater and calcifying fluid (e.g., Trotter et al., 2011).*

Author response to above comments:
We will change this information to "Moreover, due to the ability of some calcifying organisms to alter carbonate chemistry at their site of calcification, empirical species-specific calibrations between $\delta^{11}B_{CaCO_3}$ and seawater pH are likely more appropriate than theoretical α values if the goal is to reconstruct ambient seawater conditions (Anagnostou et al., 2012; Hönisch et al., 2004; Krief et al., 2010; Rae et al., 2011; Reynaud et al., 75 2004; Trotter et al., 2011).

*Section 1.1 could use greater clarity for pH terms. The manuscript starts with discussing seawater pH, and _11B as a seawater pH proxy, but transitions to calcifying fluid pH in this section. For the sake of readability, I suggest you define these separately here (seawater pH = pHsw and calcifying fluid pH = pHcf , or similar), and use throughout the text.*

*L83-85: This is true, although you could additionally cite several recent reinterpretations of boron incorporation into carbonates (Norieaux et al., 2015; Uchikawa et al., 2015, Balan et al., 2016).*

*L94-95: Suggest rewording to "organisms' ability to regulate pH at their site of calcification"*

*L108: Ca2+*

*L109-125: This would be better placed at the beginning of Section 1.2, before the discussion of OA reducing [CO2□3 ]*

*L125: Specify that K\*sp is a function of temperature*

*L136: Is there a section 1.3? If not, change this to "1.3"*

*L137-138: Rephrase to "may record" and cite studies suggesting that carbonate _11B records calcifying fluid pH. While it has been hypothesized that _11B records pH in the calcifying microenvironment, to say that _11BCaCO3 should record calcifying microenvironment pH is a stretch given current uncertainties in how pH is controlled in these microenvironments (e.g., Section 1.2), and uncertainties in the _11B proxy (see discussion in Farmer et al., 2015).*

Author response to above comments (section 1.1. to L138):
We will change this information as recommended by the reviewer. For line 125 we will specify that Ksp is a function of temperature and salinity in the text. For L137-L138, we will rephrase the sentence with "*may record*" and we will cite the following papers (McCulloch et al. 2012; Holcomb et al. 2014; Farmer et al. 2015; Martin et al. 2016)

*L164-171: Move to Methods section as a "Materials" subsection. Also, how were the specimens subsampled for isotopic analysis? Were they bulk homogenized or subsampled on particular growth features? Is there only one specimen per taxa or multiple specimens? This is very important to include as it might shed light on some of the poor reproducibility you*

*observe (especially for the coralline alga).*

Author response:
We will add a new materials section and provide a short description of how the samples were sampled. Part of the information is included in Ries et al. (2009), but we will include a short description to clarify. The samples were subsampled on new growth, homogenized, and multiple specimens per taxa were evaluated (as indicated by the name of the sample in table 3). The extent of new growth was evaluated based on the addition of a barium spike as described in Ries (2011) and this information was used to guide subsampling. We will also clarify the differences between the intra-specific (same species but different organisms), intra-organism (sub-sampling the same organism), and analytical reproducibility. As the reviewer noted, the intra-specific reproducibility for the red coralline alga is large, however, the intra-organism and analytical reproducibility is not. This suggests that there is likely geochemical heterogeneity in the carbonate matrices of this species, with large variability observed between organisms but the analytical reproducibility is robust. However this important point will be highlighted and better explained in the text, as it was also noted by Reviewer 3.

*L198-200: You can remove the sentence starting with "All samples: : :" since you dis- cuss this immediately below.*

Author response to above comments:
We will change this information as recommended by the reviewer.

*L209: Were the samples just rinsed in buffered UHQ water, or were they stored in the water? It is unclear if they were then acidified in this water medium (or not).*

Author response to above comment:
The samples were simply rinsed in the water. We will change this sentence to: "Samples were then ultrasonicated for 10 minutes, centrifuged, and then the acid was removed. The samples were washed twice with pH- buffered UHQ water, centrifuged and the water was removed.

*L224 (Batch method): Can you specify the type of microcentrifuge tubes used (polypropylene vs. PFA/Teflon), and whether the tubes were reused between samples? Or did you transfer the resin into separate microcentrifuge tubes for each sample? If the latter, how did you store the resin between samples?*

Author response to above comment:
We used polypropylene tubes and they were not used between samples. The resin was prepared following the description on lines 194-195: "The resin was crushed and sieved to a desired $100 - 200$ mesh, then cleaned and conditioned to a pH of 7 ($6.8 - 7.2$)." We then take a small aliquot of the resin and place it into the polypropylene microcentrifuge tubes, which are then washed and prepared further. See lines 225-227 "Cleaned samples (pH 7) were transferred into acid-cleaned micro-centrifuge tubes (500 μL) containing 5 mg of resin, which is B-cleaned with 500 μL of 0.5 M $HNO_3$, and then rinsed with 500 μL of MQ water (buffered to pH 7 with 2 % $NH_4OH$) three times to elute the other cations in the matrix and achieve pH 7."

For clarity, we will make the appropriate links between these two sections and indicate that they are indeed polypropylene tubes.

*L252: "The _11B was also evaluated: : :" Please explain. This reads as if you used the internal carbonate standards to correct your _11B values, which would not be appropriate. As these are internal standards, do you mean to say that you are using them to evaluate the efficacy of the preparation and measurement protocol?*

Author response to above comments:
These are not internal standards and were not used to correct our values. They are external reference standards that were analysed multiple times and were used to evaluate the method. The sentence will be changed to: "The $\delta^{11}B$ of the external calcium carbonate standards JCp-1 (Porites sp.), NEP (Porites sp.) and JCt-1 (hard clam) were also evaluated, which were processed in the same manner and are reported in the results section (see Section 3.1.1) alongside their published reference values (Foster et al., 2013; McCulloch et al., 2014)."

*L285-286: Unnecessary subsection; please remove.*

Author response to above comments:
Ok.

*L308-322: Based on Table 3, it seems that batch separation with NH3 injection was most commonly used. Does that reflect your experience with the different methods and which one seemed most replicable and user friendly? Can you make a recommendation on which separation and injection methodology you think other others should follow?*

Author response to above comments:
We do discuss this point on lines 312-317.

*L320: Can you comment on how much lower the batch method blanks were? The procedural blanks are listed as sub-nanogram (L235), but I cannot find a distinction between column vs. batch protocols.*

Author response to above comments:
We will add a comment here, but typically the column method had a blank that was near 0.5 ng and the batch method had blanks as low as 90 pg.

*L334-360: Multiple references to a phantom Figure 5. Please check Figure references throughout text.*

Author response to above comments:
We will correct this.

*L360 (Coralline red alga): Please comment on why the _11B values for these specimens are so different (_3.7 per mil uncertainty is massive!).*

Author response to above comments:
We have already responded to this comment above, see lines 164-171. In short, the standard deviation represents individual-to-individual variation (i.e. vital effects) and not the analytical precision, which is robust.

*L368-372. Couldn't both be possible-e.g., microenvironment pH adjustment and boric acid incorporation? If that was the case, could you actually determine the pH at the site of calcification? I'd strongly recommend including a figure and calculations showing how the derived value of pHcs would change as a function of varying % boric acid incorporation. Moreover, pHcs=9.4 seems pretty extreme. Is there any evidence for a physiological advantage to a calcifying organism obtaining such alkaline pH in its calcifying medium? I don't disagree with the proposed mechanism (algal photosynthesis), just the magnitude. I imagine that at this pH, CaCO3 would spontaneously precipitate (due to massively high omega), which would not be desirable for the organism. Finally, note that NMR is not useful for quantifying % boric acid incorporation (see and reference Balan et al., "First-principles study of boron speciation in calcite and aragonite" GCA 193, 2016).*

Author response to above comments:
Yes, it could be a combination of two. If partial boric acid incorporation is the only source resulting in a high boron isotopic composition in coralline red algae, the portion of boric acid would need to be extremely high (>75%), which contradicts the observations of inorganic or organic calcite (e.g. Cusack et al., 2015 estimated 30% trigonal B in the calcite lattice of a different species of coralline algae), therefore, we can not rule out the potential of pH up-regulation in this species. If 30% of the boric acid incorporated into the calcite, as suggested by Cusack et al. (2015), the calcification site pH will be still as high as 9. In addition, Short et al. (2015) observed that epiphytic turf algae can modify seawater chemistry (up to a pH of 9) within the diffusive boundary layer above coralline algal crusts, therefore a calculated pH of 9 is possible. We will expand our discussion on this section to make our arguments more clear on the points above and include a new figure (Figure 5; see attached) with the figure caption: "The influence of pH on the speciation of boron in seawater and $\delta^{11}B$ (adapted from Rollion-Bard,2011b). The solid and dashed curves represent the $\delta^{11}B$ composition that would result from the incorporation of different amounts of $B(OH)_3$ into the marine carbonates. The dashed vertical lines represent the calculated pH based on the assumption that 0% $B(OH)_3$ is incorporated into temperature coral and 0%, 30% and 75% $B(OH)_3$ is incorporated into coralline alga."

*Section 4.3: Seems out of order. I find it more intuitive to present the equation for pH calculation first (L455-460), then discuss/test the assumptions of this approach (Section 4.3.2), then finally loop around to the best estimate of pHcs and comparison to OA responses.*

Author response to above comments:
We agree, the order can be modified to be clearer.

*L426: This is not the correct terminology. This study's data do not suggest that the B isotope fractionation is species dependent; there is no direct measurement of the fractionation in this study. Rather, these data suggest that the B isotope composition of these taxa cannot be explained solely by borate incorporation at ambient seawater pH.*

Author response to above comments:
We originally used this terminology since the B isotope composition of the seawater should be consistent throughout all the conditions, but the reviewer is correct in that this was not directly measured. We can use the recommendation of the reviewer to avoid ambiguity.

*L430: Specify the assumption that only borate is incorporated here*

Author response to above comments:
We will correct this.

*L449: "by testing the factors that may influence the theoretical model of borate _11B variation as a function of pH" The theoretical model of carbonate _11B reflecting seawater pH has three parts: 1) borate _11B varies with pH in a known fashion (requires knowledge of pH, pKB\* as you say here); 2) carbonates are calcified from unmodified seawater; 3) boron in carbonates results from solely borate incorporation. Only by combining all three assumptions can you use carbonate _11B to record seawater pH. You've discussed 1; please discuss what your data suggest about 2 and 3, and how uncertainties in these assumptions could influence your data. You've already discussed 2 (modification of calcification site chemistry) throughout the discussion; bring it all together here.*

Author response to above comments:
We can modify this. We will ensure point 3 is addressed by referring to the expanded discussion on B incorporation (see response to lines 368-372).

*L475-495: The first paragraph (L475-486) is just rehashing the introduction and can be removed. The second paragraph (L487-495) is the meat of this.*

*L478: Figure 2 instead of Figure 3?*

Author response to above comments:
We will make these changes

*L499: I would urge caution with this relationship between pH elevation and OA response, as it is at best a qualitative relationship. Also, seeing as this is a central pointof the manuscript, I would recommend including a figure to illustrate the relationship between pH elevation and OA response (something like Doney et al. 2009's Figure 4 may work).*

Author response to above comments:
We agree that a simple determination of calcifying fluid  pH could not be considered a strong determinate of OA response, as ideally one would want to assess how CF responds when challenged with changing conditions. However data on OA sensitivity exists from previous work on our samples (e.g. Ries et al., 2009), therefore it made sense to include a reference and a brief discussion of that information here and in Table 4. We do not think it is necessary to include an illustration between pH elevation and OA response since this type of relationship

was already analysed for our samples in Ries et al., 2009.

*L505: Please note that this "species-specific" calibration approach is not new; it has been the standard procedure in the boron isotope community for years, as demonstrated by numerous studies that should be cited here (e.g., Sanyal et al., 1996; 2001; Hönisch et al., 2003; Trotter et al., 2011; Anagnostou et al., 2012; etc.)*

Author response to above comments:
We will cite those papers, it was an oversight on our part to not include that information.

*Figure 2: -The seawater borate curves must be mislabeled; all else being equal, increasing alpha will lead to a lower _11B-borate at lower pH. I think the dotted line should be Kakihana and the solid line should be Klochko (see also Fig. 4).*
*-Why did you choose to plot these specific data? The chosen ones seem quite random, and there are many other data out there worth considering (as your Table 5 illustrates) that may be most appropriate for comparison with the carbonates presented in this study.*
*-Is the large pH range on the x-axis (7-10) necessary? It is difficult to make out the individual studies. -Note also typos: "Hönisch", "Brachiopod" and "Penman".*
*Figures 2 and 4: Please also plot _11B – boric acid for the fractionations*

Author response to above comments:
We aim to show boron isotopic composition from some of the most studied marine biogenic carbonate archives including corals, foraminifera and bivalves. We also want to show that the data has been reported to follow different borate fractionation curves. Therefore, we have chosen studies that have more than two boron data points in a wide range of pH conditions, which aim to calibrate/validate the $^{11}$B-pH proxy in different species. For the above purpose, we will also replace the reference from Foster et al., 2008 to Sanyal et al., 1996 and Henehan et al., 2013. To incorporate all the data, the x-axis will range from 7-10. It currently shows the limits of sensitivity of the borate curve to pH, which is relevant given where the coralline algae data fall.

**Reply to reviewer 3**

We wish to thank the anonymous reviewer for their critical analysis of our manuscript and their helpful comments. We believe that we can address all of the major comments as indicated in the discussion below.

*The manuscript by Sutton et al reports the boron isotope compositions of various marine calcifyers (coralline red alga, urchins, worm, coral, oyster). All the samples came from culture experiment (T=25_C, pCO2=409 μatm) and so should record the same _11B values if no vital effects are present. The _11B range of all the data is about 20‰ and seems to show the biological control on the calcification pH. I found the data interesting, but I think that there are a lot of repetitions through the text. Even if it is mentioned in the case of coralline red alga, the influence of B3 is not really taken into account. For example, the presence of B3 was also shown in corals, and it was not described in the text. In the figures, the symbols should be different between the calcium carbonate polymorphs.*

Author response
We agree that we could expand the discussion on the influence of B3. We assume that in the case of corals, the reviewer is referring to Rollion-Bard et al. 2011b. We did not include a discussion for B incorporation in corals since, as the second reviewer pointed out *"NMR is not useful for quantifying % boric acid incorporation (see and reference Balan et al., "First-principles study of boron speciation in calcite and aragonite" GCA 193, 2016)".* Although NMR gives evidence that trigonal boron is present in the calcite lattice, it cannot determine whether boric acid was in fact incorporated or if the trigonal boron originated from borate (see alternative mechanisms of boron incorporation in for example Klotchko, 2006; Noireaux et al., 2015).

However, we agree with the reviewers that we should expand our discussion within section 4.2 to provide more information on the different factors (including seawater pH and calcifying fluid pH) that can influence the speciation of boron and $\delta^{11}B$ for inorganic calcite and aragonite. We will expand our discussion on this section to make our arguments more clear and include an extra figure (see attached Figure 5) with the following Figure caption: "The influence of pH on the speciation of boron and $\delta^{11}B$ (adapted from Rollion-Bard, 2011b). The solid and dashed curves represent the $\delta^{11}B$ composition that would result from the incorporation of different amounts of $B(OH)_3$ into the marine carbonates. The dashed vertical lines represent the calculated pH based on the assumption that 0% $B(OH)_3$ is incorporated into temperature coral and 0%, 30% and 75% $B(OH)_3$ is incorporated into coralline alga."

*-L53: B(OH)4- -L85: please mention the study of Noireaux et al (2015)*
*-L142: please mention the study of Noireaux et al (2015)*
*-L151-152: Please mention the studies of Rollion-Bard et al (2003, 2011)*
*-L154: Please mention the study of Jorgensen et al (1985)*

Author response
We will mention these studies.

*-L108, 127: "2" must be in superscript*
*-L327: "range in range", please correct*

*-L358: please remove the part of the sentence concerning boron isotopes. In this sentence, it is explained that there is an enrichment of 11B in corals and that it is supported by 'boron isotope analyses' (of course!).*
*-L477: Kakihana et al (1977) instead of Kakihana (1977)*
*-section 4.3.3: It was already mentioned, please delete this section*
*Table 3: 'JCp-1' instead of 'JCP-1'.*
*-L420: Klochko et al (2006), instead of Klochko (2009)*
*-Table S1: In the caption, specify the pKa and alpha used.*

Author response:
These will be changed

*-section 4.3.3: It was already mentioned, please delete this section*

Author response:
We think that this information is important to keep in the manuscript since it allows us to present a hypothesis that might explain the wide range of $\delta^{11}B_{CaCO3}$ (20 ‰) observed for the species evaluated in this study. In addition, we would like to add a reference to table 4 to highlight the importance of this information: "Furthermore, there appears to be a moderate inverse relationship between the species' relative ability to elevate calcification site pH and their empirically determined vulnerability to ocean acidification (Table 4)."

*-L221: Interest for what? Why the data are not shown in the manuscript?*

Author response:
The other elements (Ca, Na, Ba, U) are of interest to analyse since they can indicate whether the sample matrix has been washed out of the column. To be more clear, we will change this sentence to: Small aliquots of each sample were measured by single collector HR-ICPMS prior to analyses by MC-ICPMS to verify the retention of B on the column and removal of other elements (e.g. Ca, Na, Ba, U).

The B data are shown in the manuscript, see lines 275-278.

*-L243-244: It was already mentioned, please delete*

Author response:
I disagree, we did not state "Boron yields are evaluated by tracking B throughout the entire procedure." prior to this sentence.

*-L256: I suppose that there are older references than McCulloch et al (2014) for the MC-ICP-MS method.*

Author response:
Yes, this is true. McCulloch et al. 2014 did a great job of describing the development of the MC-ICP-MS method for the analysis of B isotope analyses and in this case we felt it useful to cite a recent paper that summarizes the state of the art on method development. We will change the citation as follows: (see McCulloch et al, 2014 for up-to-date summary of methods).

*- section 3.1.1.: Do you have an idea why the measurements on JCt-1 are more variable?*

Author response:
The JCt-1 measurements in our study were variable for the different methods of sample injection (NH3 and d-DIHEN) but we did not see this variability for other samples or standards analysed with the same methods. We are not sure why the d-DIHEN method did not provide accurate results for JCt-1, but this has not influenced our conclusions. Further, the errors are still within acceptable limits as can be seen by the variability of the inter-laboratory study (lines 280-281).

*-L288, 324: please add the errors on the _11B values*

Author response:
We didn't think it was necessary here since we are indicating the overall range in $\delta^{11}B$ values. We present the error bars related to the $\delta^{11}B$ of each species in the sentences that follow.

*-L290: Why the error on the _11B value of the coralline alga is so high?*

Author response:
As the reviewer noted, the intra-specific (same species but different organisms) reproducibility for the red coralline alga is large, however, the intra-organism (sub-sampling the same organism) and analytical reproducibility is not (see Table 3). This suggests that there is significant geochemical variability across the skeleton of this organism, but the analytical reproducibility is robust. We will make a more specific note of this in the text starting on line 289: "…and summarized in the text that follows. Note that the average data presented here (Table 4) represent intra-species reproducibility ( i.e. measured differences between individual organisms of the same species), which can be substantial however, the intra-organism (sub-sampling of same organism) and analytical reproducibility (Table 3) are typical of single organism $\delta^{11}B$ analyses."

*-L334: No, in Noireaux et al (2015) there is a clear effect of the mineralogy (see figure 1)*

Author response:
This is a glaring error on our part. We tried to simplify an argument, and the message was lost in translation. Thank you for picking up on this.

The sentence should read "Although Mavromatis et al. (2015) also found that polymorph mineralogy influences both the B/Ca ratio (higher in aragonite than calcite) and speciation of B in inorganic $CaCO_3$ (borate/boric acid ratio higher in aragonite than calcite), B incorporation alone does not appear to influence boron isotope fractionation.

*-L370-371: What would be the pH of calcification if there is effectively 30% of B3? The _11B value of coralline alga could result from the combination of a pH increase and the incorporation of a certain proportion of B3.*

Author response:

The reviewer asks an interesting question that can not be answered simply but does merit an extended discussion in the manuscript. Several related factors might influence the boron isotope composition of a calcifying organism including: the pH at the calcification site, the influence of pH on the speciation of B at the calcification site, boric acid incorporation into the calcite matrix, and the influence of boric acid on the trigonal structure of the lattice. Cusack et al. (2015) suggested that 30% of B in the calcite of a different coralline algae species was present in the trigonal B3 form; however, this does not necessarily suggest that the calcification fluid contained 30% boric acid or that 30% boric acid 70% borate was incorporated into the calcite lattice. Further empirical work is needed to clarify this relationship. However, if we were to ask the hypothetic scenario; what would be the pH at the calcification site be if 30% boric acid was available at the calcification site prior to biomineralization., we can answer that the calcification site pH would still be as high as 9, which is well above the ambient seawater pH of 8.1 (see table 4). As mentioned above, we will expand our discussion on this section to make our arguments clearer and include an extra figure (see attached Figure 5) with the following Figure caption: "The influence of pH on the speciation of boron and $\delta^{11}B$ (adapted from Rollion-Bard, 2011b). The solid and dashed curves represent the $\delta^{11}B$ composition that would result from the incorporation of different amounts of $B(OH)_3$ into the marine carbonates. The dashed vertical lines represent the calculated pH based on the assumption that 0% $B(OH)_3$ is incorporated into temperature coral and 0%, 30% and 75% $B(OH)_3$ is incorporated into coralline alga."

*-section 4.2.3: What are the calculated pH if the results of Noireaux et al (2015) for inorganic calcite are taken into account?*

Author response:
We agree with the reviewers that we should expand our discussion within section 4.2 (as described previously in this response to reviewer) to provide more information on the different factors (including pH) that can influence the speciation of boron and $\delta^{11}B$ for inorganic calcite and aragonite.

*-L402: please remove 'Notably....worm tubes'*

Author response:
We will change this to: "To our knowledge these are the first reported B isotope measurements for worm tubes"

*-L420: please remove 'Notably....oysters'*

Author response:
We will change this to: "To our knowledge these are the first reported B isotope measurements for oysters"

*-L495: It is obvious. I do not see the point here.*

Author response:
We thought it was important to clarify this point since it may not be obvious to all readers the extent to which $\alpha$ varied and our aim is to make the manuscript accessible to readers who may not all be very familiar with the boron isotope proxy so some basic statements like this can be

valuable.

*Figure 1: Please use the alpha of Klochko et al (2006) and specify in the caption the pKa used and the alpha used.*

Author response:

[Figure]

We will modify the figure attached above. The calculations are based on pKb = 8.1 (in seawater at 25 °C and a salinity of 35 under atmospheric), alpha=1.0272, d11Bsw=39.61

*Figure 2: Please add data of Reynaud et al (2004), Lécuyer et al (2002), Farmer et al (2005). Please use the full name species of the foraminifera. 'Brachiopod' instead of 'Brochiopod';*

*'Penman' instead of 'Penmen'.*

Author response:

[Figure]

········ Seawater borate    pHsw
    α = 1.0194 (Kakihana *et al.*, 1977)
──── Seawater borate
    α = 1.0272 (Klochko *et al.*, 2006)

+  *Acropora nobilis* (Hönisch *et al.*, 2004)
□  *Porites cylindrica* (Hönisch *et al.* 2004)
○  *Stylophora pistillata* (Holcomb *et al.* 2014)
▲  *Orbulina universa* (Sanyal *et al.*, 1996)
◆  *Globigerinoides  ruber* (Henehan *et al.*, 2013)
✕  Brachiopod (Penman *et al.*, 2013)

We aim to show boron isotopic composition from the most studied marine biogenic carbonate archives including corals, foraminifera and bivalves. We also want to show that the data has been reported to follow different borate fractionation curves. Therefore, we have chosen studies that have more than two boron data points in a wide range of pH conditions, which aim to calibrate/validate the $^{11}$B-pH proxy in different species. For the above purpose, we will also replace the reference from Foster et al., 2008 to Holcomb et al., 2014, Sanyal et al., 1996 and Henehan et al., 2013.

**In addition to the response to reviewers, we have also included a full listing of all relevant changes made in the manuscript (see below).**

**Changes are listed per page and line numbers refer to the marked up manuscript that follows.**

**Page 1-**
**L14-38:**
*Inserted* "boron:,
*Deleted* "of boron";
*Inserted* "($\delta^{11}$B),
*Deleted* (B),
*Inserted* "of"
*Deleted* "in"
*Inserted* "skeletal"

*Deleted* ($\delta^{11}$B)
*Inserted* "potentially critical", removed "dominant"
*Inserted* "Bates, 2007; Feely et al., 2008; Dore et al., 2009; Byrne et al., 2010; Gonzalez-Davlia et al., 2010;; Feely et al., 2016"
*Inserted* "marine"
*Deleted* "found"
*Inserted* "revealed"
*Inserted* "widely amonst"
*Deleted* "between"
*Inserted* "e.g.,"
*Inserted* "how an organism's"
*Inserted* "its specific"
*Deleted* "the various"
*Deleted* "organismal"

**Page 2**
**L39-73**
*Deleted* "Saenger et al., 2013"
*Deleted* ";  Zinke et al., 2014"
*Deleted* "in time and space"
*Inserted*  "(Zeebe and Wolf-Gladrow, 2001)"
*Inserted* "Hemming and Hanson, 1992
*Deleted* "Pagani et al., 2005"
*Deleted* "noted"
*Inserted* "expressed"
*Deleted* "NIST, Gaithersburg, MD, USA"
*Inserted* "Catanzaro et al, 1970"
*Deleted* "0.04"
*Deleted* "fractionation factor between boric acid and borate ion in seawater ($\alpha$)
*Deleted* "has been"
*Inserted* "was"

**Page 3**
**L74-102**

*Inserted* "empirical species-specific calibrations between"
*Deleted* "and/or potential isotopic fractionation during boron incorporation in biogenic carbonates, species-specific fractionation factors and transfer functions are likely more appropriate than theoretical α values
*Inserted* "and seawater pH (pHSW) are likely more appropriate than theoretical"
*Inserted* "values if the goal is to reconstruct ambient seawater conditions"

[revised manuscript text omitted]

**Page 4**
**L122-144**
*Inserted* "and is influenced by temperature and salinity"
*Deleted* "seawater"
*Inserted* "SW"
*Inserted* " (pHCF)"
*Deleted* "maintain and elevated"
*Inserted* "regulate"
*Inserted* "macro"

**Page 5**
**L162-182**
*Inserted* "CF"
*Deleted* " at the site of calcification"
*Deleted* "should"
*Inserted* "may"
*Inserted* "CF"
*Inserted* "(McCulloch et al., 2012 Holcomb et al., 2014; Farmer et al., 2015; Martin et al., 2016)"
*Inserted* "pHSW"
*Deleted* "pH of seawater"
*Inserted* "; Noireaux et al., 2015"
*Deleted* "exists because"
*Inserted* "is that"

**Page 6**
**L182-220**
*Changed* " pH" to "pH$_{CF}$"
*Changed* "seawater pH" to pH$_{SW}$
*Changed* "calcifying fluid pH" to pH$_{CF}$
*Inserted* "Rollion-Bard et al., 2003, Rollion-Bard et al., 2011b;"
*Inserted* "pHCF"
*Deleted* "calcifying fluid pH"

*Inserted* "pHSW"
*Deleted* "seawater pH"
*Inserted* "Jorgensen et al., 1985;"
*Inserted* "pHCF"
*Deleted* "extrapallial fluid pH"
*Deleted* "pH of the organisms' calcifying fluids"
*Deleted* "the organism's surrounding seawater"

[revised manuscript text omitted]

*Deleted* "organisms"

*Inserted* "species"

*Inserted* "pH$_{SW}$"

*Deleted* "seawater pH"

*Inserted* "pH$SW$"

*Deleted* "seawater pH"

**Page 11**
**L-371-409**

*Changed* "our" to "the"

*Changed* "d-DHIEN" to d-DIHEN" (throughout)

*Changed* "for" to "of"
*Changed* "calcification site pH" to "pH$_{CF}$"
*Inserted* "(i.e. abiogenic) "
*Deleted* "polymorph mineralogy was not found to influence boron isotope fractionation"
*Inserted* "B incorporation alone does not appear to influence boron isotope fractionation"
*Inserted* "(corals, serpulid worms)"
*Changed* "Fig. 5" to "Fig. 4"
*Inserted* "pHCF"
*Deleted* "in pH at the site of calcification"
*Changed* "isotope composition" to "stable isotopic compositions"
*Deleted* "calcification site pH"
*Inserted* "SW
*Inserted* "CF

**Page 12**
**L-410-449**
*Changed* "similar to" to "consistent with"
*Changed* "other literature-based values determined" to "previously published values"
*Changed* "see Figures 3 and 5" to "see Figures 2 and 4"
*Deleted* "calcifying fluid pH"
*Inserted* "; Wall et al., 2016"
*Deleted* "boron isotope analyses (e.g., Anagnostou et al., 2012; Krief et al., 2010; Trotter et al., 2011; McCulloch et al., 2012),"
*Inserted* "pH-sensitive"
*Deleted* "pH"
*Deleted* "data"
*Deleted* "value"
*Modified text to* "…, which is not consistent with prior observations for inorganic and organic calcite (e.g., Cusack et al., 2015, reported 30% trigonal B in the calcite lattice of a different species of coralline algae).Therefore, boric acid incorporation cannot alone rule out pH$_{CF}$ as a potential driver of the anomalously elevated $\delta^{11}B_{CaCO3}$ observed here for coralline algae.). Moreover, although nuclear magnetic resonance spectroscopy reveals that trigonal boron is present in the calcite lattice, it cannot determine whether boric acid was incorporated directly into the calcite lattice, or if the trigonal boron originated from borate post-mineralization (e.g., see alternative mechanisms of boron incorporation discussed in Klochko, 2006; Noireaux et al., 2015). Nevertheless, if 30% ofskeletal B is indeed directly incorporated into the calcite lattice of coralline algal skeleton, as reported by Cusack et al. (2015), pH$_{CF}$ would still need to be as high as 9 to explain the anomalously high $\delta^{11}B_{CaCO3}$ (see Fig. 5). Short et al. (2015) observed that epiphytic turf algae can increase pH$_{SW}$ up to 9 within the diffusive boundary layer above coralline algal crusts, driven by the algae's photosynthetic drawdown of aqueous CO$_2$, lending support to the idea that coralline red algae could maintain their calcifying fluid near pH 9. Thus, $\delta^{11}B_{CaCO3}$ compositions of coralline red algae may indeed reflect substantiallyelevated pH$_{CF}$ (9.4; Table 4, Fig. 4), suggesting that coralline red algae are highly efficient at removing protons and/or dissolved inorganic carbon from their calcifying medium. "

**Page 13**
**L-450-487**
*Inserted* "SW"
*Deleted* "using"
*Inserted* "from"

*Inserted* "pHCF"
*Deleted* "pH"
*Inserted* "theoretical value of $\delta^{11}B$ for seawater borate"

*Deleted* "calculated $\delta^{11}BCaCO3$"
*Deleted* "calcification site pH"
*Deleted* "pH"
*Inserted* "Note that the $\delta^{11}B$ values for these high-Mg calcite-precipitating organisms are also not consistent with significant boric acid incorporation into the carbonate lattice (Fig. 5)."
*Modified* "Serpulid worm" to "The serpulid worm"
*Deleted* "s secrete calcareous tubes with"
*Deleted* "being the only species that"
*Inserted* "their calcareous tube from a"
*Deleted* "In order to produce their calcareous tubes, *H. crucigera*"
*Inserted* "The worms initially"
*Inserted* "ultimately"
*Inserted* "SW"
*Deleted* "pH at its site of calcification"
*Deleted* "Notably,
*Inserted* "To our knowledge

*Changed* "published $\delta^{11}BCaCO3$ data" to reported B isotope measurements"
*Inserted* "and the $\delta^{11}B$ values for this mixed mineralogy precipitating organism is not consistant with significant boric acid incorporation into the carbonate lattice (Fig .5)."
*Deleted* "Mollusks, such as

**Page 14 and 15**
**L488-526 and up to L534**
*Deleted* "pH
*Inserted* "pHSW
*Inserted* "pHCF
*Deleted* "pH at the site of calcification
*Inserted* "et al.,"
*Changed* "Fig. 6 to "Fig. 5"
*Deleted* "Notably, these are the first published $\delta^{11}B_{CaCO3}$ data for"
*Inserted* "To our knowledge, these are the first reported B isotope measurements for oysters and the $\delta^{11}B$ values for this low Mg calcite-precipitating organism are not consistant with significant boric acid incorporation into the carbonate lattice (Fig. 5).
*Deleted* "4.3 What does a species' $\delta11BCaCO3$ reveal about its calcification site pH and relative sensitivity to ocean acidification?"
*Deleted* "Understanding how marine organisms calcify is a critical requirement for understanding and, ideally, predicting their physiological responses to future ocean acidification (e.g., Kleypas et al., 2006).
*Deleted* "Notably, the temperature coral (O. arbuscula) and coralline red alga (Neogoniolithion sp.) have higher calculated calcification site pH (based on their boron isotope composition) than the other organisms. As discussed above (section 4.2), one possible explanation for these differences is that corals (and potentially coralline red algae) maintain their calcifying fluids at a higher pH than the calcifying fluids of other calcifying marine organisms."
*Moved to section 4.3.3 (from 4.3)* "Notably, the different species' $\delta11BCaCO3$ and reconstructed calcification site pH appeared to exhibit a moderate, inverse relationship with their experimentally

determined vulnerability to ocean acidification (Ries et al., 2009). Species exhibiting more resilient 'parabolic' (e.g., coralline red alga) and 'threshold' (e.g., coral, tropical urchin) responses to ocean acidification generally exhibited a higher δ11BCaCO3 and, thus, calcification site pH than species exhibiting the more vulnerable 'negative' responses (e.g., oyster, serpulid worm) to ocean acidification (Table 4). The temperate urchin was the exception to this general trend, as it exhibited a relatively resilient parabolic response to ocean acidification yet maintained δ11BCaCO3 and, thus, calcification site pH close to that of seawater.

*Moved within section 4.3* "In the absence of empirical measurements of calcifying fluid temperature, salinity, and δ11B, these parameters are generally assumed to reflect seawater."

*Deleted* "However, the large variability in the calculated calcification site pH for these organisms (e.g. 7.9-9.4; Table 4) that were grown in near identical seawater conditions (pH = 8.0-8.2; Table 4) suggests that a biological process (e.g., regulation of calcification site pH) is governing boron isotope fractionation within the calcifying fluids and shells of marine calcifiers. Below,"

*Moved down to section 4.3.1* " we evaluate the sensitivity of calculating $pH_{CF}$ from measured $\delta^{11}B_{CaCO3}$ composition by testing the two principal factors that may influence the theoretical model of borate $\delta^{11}B$ variation as a function of both $pH_{CF}$ and $pH_{SW}$; namely $pK_B$ and $\alpha$. A sensitivity analysis of $\delta^{11}B$ in seawater was not conducted since all organisms evaluated in this study were exposed to seawater from the same source and, thus, of identical $\delta^{11}B$ composition.

*Deleted* "identical $\delta^{11}B$ composition."

**Page 15**
**L-535-562**
*Inserted* "4.3 Estimating $pH_{CF}$ from $\delta^{11}B_{CaCO3}$"
*Inserted* "The 6 species of calcifying marine organisms investigated in the present study exhibited average $\delta^{11}B_{CaCO3}$ compositions ranging from 16.27 ‰ to 35.09 ‰ (Table 3)
*Inserted* "In the absence of empirical measurements of calcifying fluid temperature, salinity, and $\delta^{11}B$, these parameters are assumed to reflect seawater."
*Deleted* "Notably, the temperature coral (*O. arbuscula*) and coralline red alga (*Neogoniolithion* sp.) have higher calculated $pH_{CF}$ (based on their boron isotope composition) than the other organisms."
*Moved up* "4.3.1 Sensitivity of δ11BCaCO3
*Inserted* "based calculations of $pH_{CF}$"
*Inserted* "to choice of $p_{KB}$ and $\alpha$"
*Deleted* "for estimating seawater and calcification site pH"
*Deleted* "Other factors, such as the modification of seawater pH and incorporation of boric acid by marine calcifiers, are not presented here as they are discussed previously (see Section 4.2, Fig. 5). Also note that a"
*Inserted* "A sensitivity analysis of $\delta^{11}B$ in seawater was not conducted since all organisms evaluated in this study were exposed to seawater from the same source and, thus, of identical $\delta^{11}B$ composition

**Page 16**
**L565-599**
*Inserted* "CF"
*Deleted* "and the"
*Inserted* "$\delta^{11}B$ of calcifying fluid (…CF),"
*Deleted* "of seawater"
*Inserted* "$\delta^{11}B_{CaCO3}$"

*Deleted* "B(OH)$_4^-$"

*Deleted* "$^{11}$B$_{SW}$"

*Inserted* "$^{11}$B$_{CF}$"

*Inserted* "CaCO3"

*Inserted* "inherited from $^{11}$B$_{SW}$;"

*Deleted* "$\delta^{11}$B$_{B(OH)4}$, viz."

*Deleted* "pH$_{SW}$"

*Deleted* "fluid"

*Inserted* "CF"

*Inserted* "CaCO3"

*Deleted* "B(OH)4"

*Deleted* "fluid"

*Inserted* "CF"

*Deleted* "viz. $\delta^{11}$BCaCO3,"

*Inserted* "CF"

*Changed* "has" to "with"

*Changed* "calcification site pH" to "CF"

*Deleted* "will"

*Deleted* "choice of"

*Changed* "fractionation factor ($\alpha$)" to "$\alpha$"

*Inserted* "pH-dependent"

*Deleted* "the"

*Deleted* ", and pH"

*Inserted* "et al."

*Changed* "borate" to "B(OH)$_4^-$"

*Inserted* "; Wall et al., 2016

*Changed* "borate $\delta^{11}$B-pH" to "$\delta^{11}$B$_{B(OH)4}$"

*Inserted* "pH"

*Inserted* "the differing capacities of calcifying marine species to"

*Changed* "modified" to "modify"

*Changed* "pH at the organism's site of calcification" to "pH$_{CF}$"

*Changed* "calcification site pH" to "pH$_{CF}$"

**Page 17 and 18**
**L600-639 and up to L652**
*Changed* "calcification site pH" to "pH$_{CF}$"

[revised manuscript text omitted]

**Tables and Figures**
*Changed Table 4 description to :*
"Summary of the average and standard deviation (SD) of $\delta^{11}B$ for each species (‰), calculated pH of calcifying fluid ($pH_{CF}$), pH of seawater ($pH_{SW}$) during the experimental conditions, difference between $pH_{CF}$ and $pH_{SW}$ ($\Delta pH$), calcification response to ocean acidification experiments ('OA Response'; Ries et a;., 2009), and shell/skeletal mineralogy ('HMC' = high-Mg calcite;'LMC' = low-Mg calcite;Ries et al., 2009). In most cases 3 biological replicates of each species were analyzed. 'NA' = not available, only one biological replicate analysed. **Note:** SD is calculated from measurements of different individuals of the same species and this reflect interspecimen variability. Variability arising from intra-specimen variation (reflecting variability within a single specimen) and analytical error is provided in Table 3."

*Changed Table 5 description to* "Previously published $\delta^{11}B$ analyses of biogenic marine carbonates and seawater samples"

*Changed Fig 1. In response to reviewer 3 comments. Inserted new text into figure caption 1:* "The pKb is 8.6 at 25 °C and 35 psu (Dickon, 1990), α is 1.0272 (Klochko et al., 2006), and $\delta^{11}B_{SW}$ is 39.61 (Foster et al., 2010)."

*Changed Fig. 2 in response to reviewers 2 and 3 (see reply to reviewer comments)*

*Changed figure caption 2:* "8.6152" to "8.6"
*Changed Fig. 4 in response to reviewer 2 comments (see reply to reviewer comments)*
*Changed in figure caption 4:*
"with respect to" to "as a function of"
"borate $\delta^{11}B$" to "$\delta^{11}B_{B(OH)4—}$"
*Deleted* "fractionation factors"

*Added new Figure in response to reviewer comments (Fig. 5)*

[revised manuscript text omitted]
 α. Other factors, such as the modification of seawater pH and incorporation of boric acid by marine calcifiers, are not presented here as they are discussed previously (see Section 4.2, Fig. 5). Also note that A a sensitivity analysis of δ¹¹B in seawater was not conducted since all organisms evaluated in this study were exposed to seawater from the same source and, thus, of identical δ¹¹B composition.,

**4.3.1 4.3.1 Sensitivity analysis of δ¹¹B-derived pH$_{CF}$ to pK$_B$**

**Sensitivity of δ¹¹B$_{CaCO3}$ for estimating seawater and calcification site pH**

Here, we evaluate the sensitivity of calculating calcification site pH from measured δ¹¹B$_{CaCO3}$ composition by testing the factors that may influence the theoretical model of borate δ¹¹B variation as a function of pH; namely pK$_B$ and α. A sensitivity analysis of δ¹¹B in seawater was not conducted since all organisms evaluated in this study were exposed to seawater from the same source and, thus, of identical δ¹¹B composition. we explore the sensitivity of δ¹¹B$_{CaCO3}$-derived

565 The determination of $pH_{CF}$ from $pK_B$,  $\delta^{11}B$ of calcifying fluid ($\delta^{11}B_{CF}$),  and $\delta^{11}B_{CaCO3B(OH)4}$  can be summarized with the following equation (Eq. 1):

$$pH_{CF} = pK_B - \log\left((\delta^{11}B_{SW\text{-}}{}^{11}B_{CF} - \delta^{11}B_{CaCO3B(OH)4})/(\delta^{11}B_{SW\text{-}}{}^{11}B_{CF} - (\alpha \times \delta^{11}B_{CaCO3B(OH)4}) - 1000(\alpha - 1))\right);$$
$$(1)$$

[revised manuscript text omitted]
 IThe influence of pH on the speciation of boron speciation and $\delta^{11}B$ (adapted from Rollion-Bard, 2011b). The solid and dashed curves represent the $\delta^{11}B$ composition that would result from the incorporation of different amounts of $B(OH)_3$ into the marine biogenic carbonates. The dashed vertical lines represent the calculated pH based on the assumption that 0% $B(OH)_3$ is incorporated into the temperatecure coral skeleton and 0%, 30% and 75% $B(OH)_3$ is incorporated into the coralline algal skeleton. Of the calcite species examined, only the corraline algae has a $\delta^{11}B$ composition that could conceivably orginate at least in part from $B(OH)_3$ incorporation.

**Author Contribution**

 RAE and JBR _conceived of the project and_ wrote the proposals that funded the work. JBR cultured the organisms. RAE, JNS, and JBR contributed to experimental design. JNS, Y-WL, MG, EP, and RAE contributed to method development. JNS performed the measurements with assistance from EP. JNS conducted the data analysis. Interpretation was led by JNS and RAE with input from JBR and Y-WL. JNS drafted the paper, which was edited by all authors.

**Acknowledgements**

This work was supported by the "Laboratoire d'Excellence" LabexMER (ANR-10-LABX-19) and co-funded by a grant from the French government under the program "Investissements d'Avenir", and by a grant from the Regional Council of Brittany (SAD programme). RAE and JBR also acknowledge support from National Science Foundation grants OCE-1437166 and OCE-1437371. We thank J.-P. D'Olivo and the members of the UWA lab for supplying us with an aliquot of the NEP standard.

---

## Referee Report (RR1)

Jesse Farmer

Overview: Excellent job revising by the authors. This is now very close to publication quality. I have one major and several minor (mostly copyediting) comments. I recommend this for publication once the major comment on the introduction is addressed.

Big stuff:

Check table #s; they do not correspond to their order in the text.

L44-48: I think the presentation would be clearer if this section is moved down to the paragraph starting on line 76. At that point you have introduced the relevant factors ($\alpha$, $pK_B^*$) in lines 49-75.

L71-85 (and including L44-48): As in the previous version, there is still a misrepresentation of how the community gets between carbonate $\delta^{11}B$ measured on a mass spectrometer and pH. You actually get this right at the end of the discussion (L504-509), but this needs to be in the introduction and not the discussion. I will revoke any copyright claim to this and recommend the authors include the following:

First, there is the theoretical model for $\delta^{11}B_{borate}$ varying with $pH_{sw}$, given by the following equation (please include and cite Zeebe and Wolf-Gladrow, 2001):

$$pH = pK_B^* - \log\left(\frac{\delta^{11}B_{CaCO_3} - \delta^{11}B_{sw}}{\delta^{11}B_{sw} - \alpha_B * \delta^{11}B_{CaCO_3} - \varepsilon_B}\right)$$

Uncertainties in this model stem from uncertainties in the variables $pK_B^*$, $\delta^{11}B_{sw}$, and primarily $\alpha$ (as you discussed on L59-71 and evaluate in the discussion). Pagani et al. (2005), Foster and Rae (2016) have nice discussions of these uncertainties that you can cite here.

2) Using $\delta^{11}B_{CaCO3}$ to derive $pH_{sw}$ requires knowledge of the relationship between $\delta^{11}B_{CaCO3}$ and $\delta^{11}B_{borate}$. One option is to assume that $\delta^{11}B_{CaCO3} = \delta^{11}B_{borate}$, as was done by early studies (e.g., Hemming and Hanson, 1992). However, Sanyal et al. (2000-GCA and 2001-Paleoceanography) pointed out that different carbonates (inorganic calcite, *O. universa*, and *G. sacculifer*) exhibited offset relationships to $\delta^{11}B_{borate}$. To quote from the 2001 Paleoceanography paper (emphasis added):

> "It is noteworthy that empirical $\delta^{11}B$ versus pH curves for both biogenic and inorganic calcite plot close to the calculated B(OH)$_4^-$ curve, indicating that the charged species is preferentially incorporated into the carbonates. A parallel offset, however, was identified between the theoretical B(OH)$_4^-$ curve and the empirical $\delta^{11}B$ vs. pH curve of both *O. universa* and inorganic carbonates. This

suggests that the calculated $B(OH)_4^-$ curve cannot be directly applied to estimate paleo-pH from the $\delta^{11}B$ of all carbonates."

Thus, it was known that $\delta^{11}B_{CaCO3} \neq \delta^{11}B_{borate}$ even before Pagani et al. (2005). As a result, empirical calibrations between $\delta^{11}B_{CaCO3}$ and $\delta^{11}B_{borate}$ are needed to calculate pH, as discussed by Hönisch et al. (2007-commment on Pagani et al. 2005), Foster (2008), Henehan et al. (2013), Farmer et al. (2015), Henehan et al. (2016), and Foster and Rae (2016). The current practice of the field is to use these empirical calibrations to calculate $\delta^{11}B_{borate}$ from $\delta^{11}B_{CaCO3}$, and then use the above theoretical model (with its uncertainties) to calculate $pH_{sw}$. In other words, $\delta^{11}B$-based paleo-pH reconstructions do not assume that $\delta^{11}B_{CaCO3} = \delta^{11}B_{borate}$.

This leads in nicely to your Section 1.3 (L138) because your study interrogates one reason why $\delta^{11}B_{CaCO3}$ may not equal $\delta^{11}B_{borate}$ (namely, $pH_{cf} \neq pH_{sw}$).

In addition, this has one small and one big implication for the manuscript. First, on L71-75 you state "empirical species-specific calibrations between $\delta^{11}B_{CaCO3}$ and $pH_{sw}$ are likely more appropriate than theoretical $\alpha$ values". This is not correct! The empirical species-specific calibrations are presented as between $\delta^{11}B_{CaCO3}$ and $\delta^{11}B_{borate}$. To get from $\delta^{11}B_{borate}$ to pH requires the theoretical model (equation above), which depends on $\alpha$. See/cite equation 1 in Hönisch et al. (2007) and equations 24 and 26 in Foster and Rae (2016). To summarize, empirical calibrations are not more appropriate than theoretical $\alpha$ values; in fact they require $\alpha$, but they are more appropriate than assuming that $\delta^{11}B_{CaCO3} = \delta^{11}B_{borate}$.

Second, because this manuscript presents $\delta^{11}B_{CaCO3}$ values at only a single pH, you do not know the true relationship between $\delta^{11}B_{CaCO3}$ and $\delta^{11}B_{borate}$ for your calcifiers. Therefore your calculated pH values must assume that $\delta^{11}B_{CaCO3} = \delta^{11}B_{borate}$. If this study attempted to reconstruct paleo-pH, this assumption would invalidate the reconstruction and the pH values should be treated with suspicion. However, because you do not have a range of pH values for each specimen, you are essentially forced to assume that $\delta^{11}B_{CaCO3} = \delta^{11}B_{borate}$. That is OK for the purpose of exploratory studies like this one, just be sure to state this assumption explicitly in the text (see my comment on L327). In general, you have done well to highlight this assumption throughout the discussion.

Specific changes:

L108: remove "forecasted," atmospheric $pCO_2$ has definitely risen!

L114: change "that they need" to "needed"

L170-171: Specify what efforts were made to minimize sample exposure to laboratory air. Did you take the caps off just before analysis?

L187: As this is the first mention of a table in the text, this should be Table 1.

Sections 2.3 and 2.4: You use MQ and UHQ interchangeably throughout; please change all to UHQ (or MQ).

L210-211: You mention this later on L228 how the UHQ water was pH buffered; please specify that detail here as this is the first mention of pH buffered water.

L265: Briefly summarize why d-DIHEN reduces memory effects, as you do for the ammonia injection.

L282: Typo- NH3 subscript should be on $\delta^{11}$B

L280-285: For comparison of the standard values, you should do a formal statistical test (t-test) for differences of averages. The d-DIHEN $\delta^{11}$B values for JCt are just barely within overlapping 2sd of the Gutjahr values and are probably significantly different with an alpha of 0.05.

L312: Again, a formal statistical test is needed here to support this assertion.

L327: Here you should say something like "Because our specimens come from only a single pH, we cannot constrain whether the relationship between $\delta^{11}B_{CaCO3}$ and $\delta^{11}B_{borate}$ significantly differs from unity, as is observed in other marine calcifiers (refs or cite intro section). Therefore, we assume that $\delta^{11}B_{CaCO3}$ reflects only $\delta^{11}B_{borate}$, and thus that only borate ion is being incorporated into $CaCO_3$. Given this assumption, the wide variation in $\delta^{11}B_{CaCO3}$ ..."

L331-343: Two items of note here:
1) It is worth mentioning that Mavromatis and Noireaux experiments are from solutions with quite different chemistry than seawater, and thus the appropriateness of their conclusions for marine carbonates are still uncertain.
2) The B speciation from these experiments was derived via NMR. However, NMR only tells coordination state (tetrahedral/trigonal). Because there are multiple possible B incorporation pathways and B coordination (see/cite Balan et al., 2016), NMR cannot distinguish between boric acid and borate. On L 335, I would rephrase this to say "and coordination of B in inorganic $CaCO_3$ (tetrahedral/trigonal ratio higher in aragonite than in calcite)". Then say that if coordination reflected the borate/boric acid ratio, aragonite-producing species should have a universally lower $\delta^{11}$B than calcite-producing species because $\delta^{11}B_{borate}$ is always lower than $\delta^{11}B_{boric\ acid}$.

L367: Same point on NMR being coordination; at best this *may* represent boric acid incorporation.

L373-376: There it is! Excellent, move this discussion up to L331-343 area when you first discuss Mavromatis/Noireaux data, and then reference it again here.

Section 4.2.2 There are multiple instances where spaces are needed between words and at ends of sentences—check this.

Section 4.2.3 Interesting. Without tooting my own horn here, I'd recommend mentioning that the relative $\delta^{11}B$ deviations are quite similar to that observed in the high-Mg calcite of bamboo corals by Farmer et al. 2015. Perhaps there is something systematic about $\delta^{11}B$ in HMC?

L404: space between "initially" and "produce"

L425: "which has been attributed to"

L504-509: Here you say what you should in the introduction. Remove this from here and incorporate this into the intro.

Fig. 1 caption: note typo on $pK_B$

Fig. 2: please mention why you chose these particular datasets to show; there are a lot more B isotope data available than just this figure. Also, please narrow the y-axis to between 10 and 40 per mil.

Fig. 5: Nice figure! Could you also draw a line for the oyster and temperate urchin back to $\delta^{11}B_{borate}$, and then down to pH?

---

## Author Response (AR2)

We would again like to thank Dr. Jesse Farmer for their thorough review of our manuscript and their helpful comments. We believe that we can address all of the major comments indicated by Dr. Farmer as indicated in the discussion below. Note that the line numbers we refer to in our response to reviewer are from the original resubmitted manuscript (bold) or from the revised manuscript that contains the tracked changes.

**Check table #s; they do not correspond to their order in the text.**

We have checked over the table numbers and their order in the text. As a result, we have removed one reference to Table 3 on line 248. Here, we changed the text to indicate that there were at least 3 specimens per species evaluated and therefore, the use of the table reference was no longer needed.

**L44-48**: *I think the presentation would be clearer if this section is moved down to the paragraph starting on line 76. At that point you have introduced the relevant factors (α, pKB\*) in lines 49-75.*

We have made the changes recommended by the reviewer.

**L71-85 (and including L44-48):** *As in the previous version, there is still a misrepresentation of how the community gets between carbonate δ11B measured on a mass spectrometer and pH. You actually get this right at the end of the discussion (L504-509), but this needs to be in the introduction and not the* discussion. *I will revoke any copyright claim to this and recommend the authors include the following:*

*First, there is the theoretical model for δ11Bborate varying with pHsw, given by the following equation (please include and cite Zeebe and Wolf-Gladrow, 2001):*

$$pH = pK_B^* - \log\left( \frac{\delta^{11}B_{CaCO_3} - \delta^{11}B_{sw}}{\delta^{11}B_{sw} - \alpha_B * \delta^{11}B_{CaCO_3} - \varepsilon_B} \right)$$

*Uncertainties in this model stem from uncertainties in the variables pKB\*, δ11Bsw, and primarily α (as you discussed on L59-71 and evaluate in the discussion). Pagani et al. (2005), Foster and Rae (2016) have nice discussions of these uncertainties that you can cite here.*

*2) Using δ11BCaCO3 to derive pHsw requires knowledge of the relationship between δ11BCaCO3 and δ11Bborate. One option is to assume that δ11BCaCO3 =δ11Bborate, as was done by early studies (e.g., Hemming and Hanson, 1992). However, Sanyal et al. (2000-GCA and 2001-Paleoceanography) pointed out that different carbonates (inorganic calcite, O. universa, and G. sacculifer) exhibited offset relationships to δ11Bborate. To quote from the 2001 Paleoceanography paper (emphasis added): "It is noteworthy that empirical δ11B versus pH curves for both biogenic and inorganic calcite plot close to the calculated B(OH)4- curve, indicating that the charged species is preferentially incorporated into the carbonates. A parallel offset, however, was identified between the theoretical B(OH)4-*

*curve and the empirical δ11B vs. pH curve of both O. universa and inorganic carbonates. This suggests that the calculated B(OH)4- curve cannot be directly applied to estimate paleo-pH from the δ11B of all carbonates."*

*Thus, it was known that δ11BCaCO3 ≠ δ11Bborate even before Pagani et al. (2005). As a result, empirical calibrations between δ11BCaCO3 and δ11Bborate are needed to calculate pH, as discussed by H.nisch et al. (2007-commment on Pagani et al. 2005), Foster (2008), Henehan et al. (2013), Farmer et al. (2015), Henehan et al. (2016), and Foster and Rae (2016). The current practice of the field is to use these empirical calibrations to calculate δ11Bborate from δ11BCaCO3, and then use the above theoretical model (with its uncertainties) to calculate pHsw. In other words, δ11B-based paleopH reconstructions do not assume that δ11BCaCO3 =δ11Bborate.*

*This leads in nicely to your Section 1.3 (L138) because your study interrogates one reason why δ11BCaCO3 may not equal δ11Bborate (namely, pHcf ≠ pHsw). In addition, this has one small and one big implication for the manuscript. First, on L71-75 you state "empirical species-specific calibrations between δ11BCaCO3 and pHsw are likely more appropriate than theoretical α values". This is not correct! The empirical species-specific calibrations are presented as between δ11BCaCO3 and δ11Bborate. To get from δ11Bborate to pH requires the theoretical model (equation above), which depends on α. See/cite equation 1 in H.nisch et al. (2007) and equations 24 and 26 in Foster and Rae (2016). To summarize, empirical calibrations are not more appropriate than theoretical α values; in fact they require α, but they are more appropriate than assuming that δ11BCaCO3 =δ11Bborate.*

*Second, because this manuscript presents δ11BCaCO3 values at only a single pH, you do not know the true relationship between δ11BCaCO3 and δ11Bborate for your calcifiers. Therefore your calculated pH values must assume that δ11BCaCO3 =δ11Bborate. If this study attempted to reconstruct paleo-pH, this assumption would invalidate the reconstruction and the pH values should be treated with suspicion. However, because you do not have a range of pH values for each specimen, you are essentially forced to assume that δ11BCaCO3 =δ11Bborate. That is OK for the purpose of exploratory studies like this one, just be sure to state this assumption explicitly in the text (see my comment on L327). In general, you have done well to highlight this assumption throughout the discussion.*

We have made some changes to this section as suggested by the reviewer, although we have not done a copy/paste of the text in the reviewer's comments, as suggested by the reviewer. Alternatively, to clarify our discussion points for this section we have modified some of the text, as the reviewer suggested, to:

(1) include the presentation of the model by Zeebe and Wolf-Gladrow (2001) and to discuss the uncertainties of this model starting at line 100.

(2) we have added the following text to address his second point starting on line 94:

"paleo-seawater pH may not simply be reconstructed by projecting measured $\delta^{11}$B of calcium carbonate ($\delta^{11}$B$_{CaCO3}$) onto a theoretical seawater borate $\delta^{11}$B ($\delta^{11}$B$_{B(OH)4}$-)- pH curve (see also Anagnostou et al., 2012; Honïsch et al., 2003; Sanyal et al., 1996, Sanyal et al., 2001; Trotter et al., 2011). Instead, the species used for paleo-seawater

pH reconstructions may require calibration through controlled laboratory experiments and/or core-top calibrations that empirically define the species-specific relationship between seawater pH ($pH_{SW}$) and $\delta^{11}B_{CaCO3}$."

And starting on line 107

"Application of this proxy also assumes that $\delta^{11}B_{CaCO3}$ reflects seawater $\delta^{11}B_{B(OH)4-}$ and, thus, seawater pH (Hemming and Hanson, 1992). Although early studies assumed that $\delta^{11}B_{CaCO3}$ was indeed equivalent to seawater $\delta^{11}B_{B(OH)4-}$ (e.g., Hemming and Hanson, 1992), Sanyal et al. (2000, 2001) observed that empirically derived $\delta^{11}B_{CaCO3}$-pH curves of biogenic and abiogenic calcites were parallel but vertically offset from the theoretical $\delta^{11}B_{B(OH)4-}$-pH curve, which led them to conclude that paleo-seawater pH cannot always be directly calculated from $\delta^{11}B_{CaCO3}$ using the theoretical $\delta^{11}B_{B(OH)4-}$-pH relationship (i.e., $\delta^{11}B_{CaCO3}$-pH relationships must be empirically calibrated for the species hosting the paleo-pH proxy).

We hope that the reviewer is satisfied with the changes that we have made.

**L108:** *remove "forecasted," atmospheric pCO2 has definitely risen!*

We have made the changes recommended by the reviewer.

**L114:** *change "that they need" to "needed"*

We have made the changes recommended by the reviewer.

**L170-171:** *Specify what efforts were made to minimize sample exposure to laboratory air. Did you take the caps off just before analysis?*

We have made the changes recommended by the reviewer. The sentence has now been changed to "Efforts were made to minimize sample exposure to laboratory air by, for example, removing caps of sample vials only when reagents were added to the samples and just prior to sample analysis."

**L187:** *As this is the first mention of a table in the text, this should be Table 1.*

We have made the changes recommended by the reviewer (see above for table order).

**Sections 2.3 and 2.4:** *You use MQ and UHQ interchangeably throughout; please change all to UHQ (or MQ).*

We have made the changes recommended by the reviewer (changed all to UHQ)

***L210-211:*** *You mention this later on L228 how the UHQ water was pH buffered; please specify that detail here as this is the first mention of pH buffered water.*

We have made the changes recommended by the reviewer.

***L265:*** *Briefly summarize why d-DIHEN reduces memory effects, as you do for the ammonia injection.*

We have made the changes recommended by the reviewer on line 326.

***L282:*** *Typo- NH3 subscript should be on δ11B*

We have made the changes recommended by the reviewer.

***L280-285:*** *For comparison of the standard values, you should do a formal statistical test (t-test) for differences of averages. The d-DIHEN δ11B values for JCt are just barely within overlapping 2sd of the Gutjahr values and are probably significantly different with an alpha of 0.05.*

We have run a t-test comparing the samples to the inter-laboratory calibration study. Using a Ho of 16.98 for the samples by Gutjahr et al. (2014), the maximum end of the values for the inter-laboratory calibration study, the d-DIHEN values for JCt are *not* significant at a significance level of p<0.05 (p-value in this case is >0.06). However, while running the t-test, we did notice that the number of samples in Table 3 should have been 6 (instead of 12, and the 2SD should have been 0.69, instead of 0.6). We have changed these numbers in the table and in the text (and double checked that no other error was made). Note, that Gutjahr et al. (2014) also identifies that JCt did not reproduce as well as the JCp standard.

***L312:*** *Again, a formal statistical test is needed here to support this assertion.*

See point above, although we will note in the text that significance was evaluated to be p>0.05.

***L327:*** *Here you should say something like "Because our specimens come from only a single pH, we cannot constrain whether the relationship between δ11BCaCO3 and δ11Bborate significantly differs from unity, as is observed in other marine calcifiers (refs or cite intro section). Therefore, we assume that δ11BCaCO3 reflects only δ11Bborate, and thus that only borate ion is being incorporated into CaCO3. Given this assumption, the wide variation in δ11BCaCO3 …"*

We have made the changes recommended by the reviewer. The text will now read as, after line 389: "…despite exposure of all species to approximately equivalent $pH_{SW}$ of 8

(see Table 4). We cannot constrain whether the relationship between $\delta^{11}B_{CaCO3}$ and $\delta^{11}B$ of borate significantly differs from unity in this experiment with a single $pH_{SW}$, because $\delta^{11}B_{B(OH)4-}$ at the species' sites of calcification cannot be measured or calculated from the data at hand, it cannot be directly compared with the measured $\delta^{11}B_{CaCO3}$ to determine if $\delta^{11}B_{CaCO3}$ necessarily reflects calcifying fluid $\delta^{11}B_{B(OH)4-}$ and, thus, $pH_{CF}$. Assuming that only the borate ion is incorporated into biogenic $CaCO_3$ (i.e., $\delta^{11}B_{CaCO3}$ = calcifying fluid $\delta^{11}B_{B(OH)4-}$), the wide variation in $\delta^{11}B_{CaCO3}$ (*ca.* 20 ‰) amongst the investigated species reared under equivalent thermo-chemical conditions may indeed arise from inherent differences in $pH_{CF}$ amongst the species."

*L331-343:* *Two items of note here:*

*1) It is worth mentioning that Mavromatis and Noireaux experiments are from solutions with quite different chemistry than seawater, and thus the appropriateness of their conclusions for marine carbonates are still uncertain.*

*2) The B speciation from these experiments was derived via NMR. However, NMR only tells coordination state (tetrahedral/trigonal). Because there are multiple possible B incorporation pathways and B coordination (see/cite Balan et al., 2016), NMR cannot distinguish between boric acid and borate. On L 335, I would rephrase this to say "and coordination of B in inorganic CaCO3 (tetrahedral/trigonal ratio higher in aragonite than in calcite)". Then say that if coordination reflected the borate/boric acid ratio, aragonite-producing species should have a universally lower δ11B than calcite-producing species because δ11Bborate is always lower than δ11Bboric acid.*

We have made the changes recommended by the reviewer. The text now reads as:
  (1) Starting on line 406: "It should also be noted that these experiments (Mavromatis et al., 2015; Noireaux et al., 2015) analyzed carbonates precipitated from non-seawater solutions; therefore, further work is needed to determine the applicability of these findings to marine carbonates."
  (2) Starting on line 404: "…and coordination of B in inorganic $CaCO_3$ (tetrahedral/trigonal ratio higher in aragonite than in calcite), B/Ca ratio alone does not appear to influence boron isotope fractionation in $CaCO_3$ (Noireaux et al., 2015)."

*L367:* *Same point on NMR being coordination; at best this may represent boric acid Incorporation AND* *L373-376:* *There it is! Excellent, move this discussion up to L331-343 area when you first discuss Mavromatis/Noireaux data, and then reference it again here.*

Although we agree with the statements made by the reviewer, we believe that the changes that we made above appropriately highlight the importance of the general discussion on **L373-376** and do not feel it is necessary to move the discussion up. We believe that the discussion is more relevant in the section on Coralline red alga.

*Section 4.2.2* *: There are multiple instances where spaces are needed between words and at ends of sentences—check this.*

We have made the changes recommended by the reviewer.

***Section 4.2.3****: Interesting. Without tooting my own horn here, I'd recommend mentioning that the relative δ11B deviations are quite similar to that observed in the high-Mg calcite of bamboo corals by Farmer et al. 2015. Perhaps there is something systematic about δ11B in HMC?*

We find the comment made by the reviewer to be intriguing, however, we do not think we have enough data to provide a solid argument suggesting a potential systematic relationship between $\delta^{11}B$ in HMC organisms, as the reviewer suggested. Especially considering that our calcareous red algae and serpulid worms are also HMC.

***L404:*** *space between "initially" and "produce"*

We have made the changes recommended by the reviewer.

***L425:*** *"which has been attributed to"*

We have made the changes recommended by the reviewer.

***L504-509****: Here you say what you should in the introduction. Remove this from here and incorporate this into the intro.*

We have made the changes recommended by the reviewer.

***Fig. 1*** *caption: note typo on pKB*

We have made the changes recommended by the reviewer.

***Fig. 2:*** *please mention why you chose these particular datasets to show; there are a lot more B isotope data available than just this figure. Also, please narrow the y-axis to between 10 and 40 per mil.*

We wanted to show the boron isotopic composition for some of the most studied marine biogenic carbonate archives including corals, foraminifera and bivalves. We also wanted to show that the data has been reported to follow different borate fractionation curves. Therefore, we chose studies that have more than two boron data points in a wide range of pH conditions, which aim to calibrate/validate the 11B-pH proxy in different species. To clarify, we will add this explanation to the figure caption for Fig. 2.

***Fig. 5****: Nice figure! Could you also draw a line for the oyster and temperate urchin back to δ11Bborate, and then down to pH?*

We have made the changes recommended by the reviewer.

**In addition to the response to reviewers, we have also included a listing of all relevant changes made in the manuscript (see below). Changes are listed per page and line numbers refer to the marked up manuscript that follows.**

**Page 1**

Line 1 - added "divergent" to title

Lines 5-15 - changed contact details including corresponding email for Jill Sutton ([Jill.Sutton@univ-brest.fr](mailto:Jill.Sutton@univ-brest.fr))

Lines 17-41 - changed order of sentences and language in abstract

**Pages 2-3**

Lines 45-60 – improved language for these sections and order of citations using the Biogeosciences format.

Lines 61-65 – moved text based on reviewer's request regarding *L44-48* (see reply to reviewer comments). Some of the text was also improved for language.

Lines 66-91 – improved language for these paragraphs

**Pages 3-4**

Lines 91-94 – deleted and included parts of this section to lines 94-99. This was done in again in response to the reviewer's request for *L44-48* (see above).

Lines 94-136 – see reply to reviewer comments regarding *L71-85 (and including L44-48).*

Lines 137-149 – Changed order of citations using the Biogeosciences format.

**Pages 5-6**

Lines 150-220 – improved language for these sections and order of citations using the Biogeosciences format.

**Pages 7-9**

Lines 227-228 – see reply to reviewer comment regarding *L170-171*

Lines 239-324 – improved language for these sections and order of citations using the Biogeosciences format.

Lines 324-328 – see reply to reviewer comment regarding *L265*

**Pages 10-12**

Lines 330-390 - improved language for these sections

Lines 390-399 - see reply to reviewer comment regarding *L327*

Lines 400-425 - improved language for these sections

Lines 426-440 – changed order of presentation by moving the Coralline red algae before the corals. Also added the scientific name of the organisms. We also added a little more detail on Coralline red algae (Lines 434-439).

**Pages 13-18**

Lines 442-544 - improved language for these sections and order of citations using the Biogeosciences format.

Lines 545-582 – We found this section to not be very clear and so we re-worked this section.

Lines 583-652 - improved language for these sections and order of citations using the Biogeosciences format.

**Tables and Figures and author contributions**

Made changes based on recommendations made by reviewer. Also made a few minor changes to Tables 1 and 3, Figure captions, and the author contribution section to improve the language of the manuscript. Figures 2 and 5 were revised and are attached to this document.

[revised manuscript text omitted]
 4) even though ourdespite exposure of all species  specimens were all exposed to a single pH condition of singleapproximately equivalent pH$_{SW}$ conditionvalues of --8 (see Table 4). As such, weWe cannot constrain whether the relationship between $\delta^{11}B_{CaCO3}$ and $\delta^{11}$B of borate significantly differs from unity in this experiment with a single pH$_{SW}$, as is observed in other marine calcifiers (see Introduction). ThereforeTherefore, we assume that $\delta^{11}B_{CaCO3}$ reflects only $\delta^{11}$B of borate, and thus that only borate ion is being incorporated into CaCO$_3$. Given this assumption, the wide variation in $\delta^{11}B_{CaCO3}$. 
[revised manuscript text omitted]